# Transcription-dependent targeting of Hda1C to hyperactive genes mediates H4-specific deacetylation in yeast

So Dam Ha[1,5], Seokjin Ham[2,5], Min Young Kim[1,5], Ji Hyun Kim[1], Insoon Jang [3], Bo Bae Lee[1], Min Kyung Lee[1], Jin-Taek Hwang[4], Tae-Young Roh [2,3] & TaeSoo Kim [1]

In yeast, Hda1 histone deacetylase complex (Hda1C) preferentially deacetylates histones H3 and H2B, and functionally interacts with Tup1 to repress transcription. However, previous studies identified global increases in histone H4 acetylation in cells lacking Hda1, a component of Hda1C. Here, we find that Hda1C binds to hyperactive genes, likely via the interaction between the Arb2 domain of Hda1 and RNA polymerase II. Additionally, we report that Hda1C specifically deacetylates H4, but not H3, at hyperactive genes to partially inhibit elongation. This role is contrast to that of the Set2–Rpd3S pathway deacetylating histones at infrequently transcribed genes. We also find that Hda1C deacetylates H3 at inactive genes to delay the kinetics of gene induction. Therefore, in addition to fine-tuning of transcriptional response via H3-specific deacetylation, Hda1C may modulate elongation by specifically deacetylating H4 at highly transcribed regions.

[1] Department of Life Science and the Research Center for Cellular Homeostasis, Ewha Womans University, Seoul 03760, Korea. [2] Department of Life Sciences, Pohang University of Science and Technology (POSTECH), Pohang 37673, Korea. [3] Division of Integrative Biosciences and Biotechnology, Pohang University of Science and Technology (POSTECH), Pohang 37673, Korea. [4] Korea Food Research Institute, Wanju 55365, Korea. [5] These authors equally contributed: So Dam Ha, Seokjin Ham, and Min Young Kim. Correspondence and requests for materials should be addressed to T.-Y.R. (email: tyroh@postech.edu) or to T.K. (email: tskim@ewha.ac.kr)

Eukaryotic transcription is regulated by covalent modifications of histones, including acetylation, methylation, phosphorylation, and ubiquitination[1,2]. Histone acetylation directly activates RNA polymerase II (RNA Pol II) transcription by disrupting the interaction between histones and DNA, thereby opening the chromatin structure, and/or recruiting factors that affect local chromatin structure and transcription. By contrast, histone deacetylation directly represses transcription.

The opposing activities of histone acetyltransferases (HATs) and histone deacetylases (HDACs) dynamically regulate histone acetylation and transcription. In yeast, multiple HATs, including SAGA, NuA3, and NuA4, are responsible for acetylation at promoter regions[3], whereas histone deacetylation is mainly mediated by the Rpd3 large (Rpd3L) HDAC[3–5]. Histone acetylation/deacetylation within coding regions affects RNA Pol II elongation, transcription initiation from cryptic promoters, and histone exchange[2]. Furthermore, co-transcriptional histone methylations at H3K4 and H3K36 play important roles in the regulation of histone acetylation. H3K4me3 at promoter regions acts as a binding site for multiple HATs and HDACs[3]. In addition, H3K4me2 targets the Set3 HDAC to deacetylate histones located at the 5′ ends of genes[6]. In 3′-transcribed regions, H3K36me2/3 enhances deacetylation by recruiting the Rpd3 small (Rpd3S) HDAC[7,8].

Hda1C includes three subunits: Hda1, Hda2, and Hda3 (ref. [9,10]). Hda1 is a class II HDAC that contains a HDAC domain and an Arb2 domain at its N terminus and C terminus, respectively[11]. Hda2 and Hda3 are structurally similar and form a heterodimeric complex; both subunits contain a potential DNA-binding domain and a coiled-coil domain at their N terminus and C terminus, respectively[11]. The coiled-coil domains of Hda2 and Hda3 are essential for both the interaction with Hda1 and the HDAC activity of the complex[11]. Notably, the Arb2 domain of Hda1 and the DNA-binding domains of Hda2 and Hda3 are not required for HDAC activity, suggesting that they may have a role in targeting of the complex to chromatin. Indeed, a recent study revealed that the Arb2 domain of Hda1 binds directly to nucleosomes in vitro[12]. Furthermore, the DNA-binding domains of Hda2 and Hda3 are able to bind to double- or single-stranded DNA in vitro[11]; however, their functions in vivo remain unknown.

Hda1C deacetylates histone H3 and H2B and interacts with the general corepressor Tup1 to repress transcription[13]. At some inactive promoters, Hda1C mutants show hyperacetylation of H3 and H2B, but not H4 or H2A[13,14]. However, a previous study showed global increases in the acetylation of H4, as well as H3, in cells lacking HDA1 (ref. [10]). In addition, HDAC activity of the Hda1C complex toward histone H4 has also been reported in vitro, suggesting that Hda1C may play important roles in H4 deacetylation[9].

In this study, we show that Hda1C binds to hyperactive genes, likely through an interaction with RNA Pol II. Surprisingly, we find that, unlike at inactive genes, Hda1C specifically deacetylates histone H4 within hyperactive coding regions. Furthermore, loss of Hda1C slightly suppresses the growth defect caused by loss of Bur1, suggesting that Hda1C may play a negative role in elongation. A comparative analysis of acetylation changes suggests that Hda1C and the Set2–Rpd3S pathway have overlapping but distinct functions in the deacetylation of histone H4 within coding regions. Finally, we find that Hda1C preferentially deacetylates histone H3 at less active genes to delay the kinetics of gene induction upon environmental changes. We propose that Hda1C has two distinct functions in transcription: (1) H4 deacetylation at hyperactive coding regions to regulate elongation, and (2) H3 deacetylation at inducible genes to fine-tune the kinetics of gene activation.

## Results

**Hda1C deacetylates histone H4 at hyperactive coding regions.** To understand the exact function of Hda1C in more detail, the acetylation patterns of histones H3 and H4 were determined by chromatin immunoprecipitation (ChIP) assays using antibodies recognizing di-acetyl H3 (K9 and K14) and tetra-acetyl H4, respectively. To determine whether Hda1C differentially affects histone acetylation depending on the transcriptional status of GAL genes, chromatin was prepared from wild-type and hda1Δ cells grown in medium containing glucose or galactose. Levels of histone acetylation were normalized to total histone content, as determined by H3 occupancy. Consistent with previous reports, H3 acetylation, but not H4 acetylation, was higher at GAL3 promoters in hda1Δ cells than wild-type cells under inactive conditions (YP-Glucose) (Fig. 1a)[13,14]. Under the same conditions, H3 acetylation within the GAL3-coding region was also increased following deletion of HDA1, suggesting that Hda1C-mediated deacetylation of H3 is not just restricted to inactive promoters. Notably, the effect of HDA1 deletion on H3 acetylation was significantly reduced when GAL3 was activated by growth in medium containing galactose (Fig. 1a). However, loss of Hda1 resulted in a strong increase in H4 acetylation within the GAL3-coding region under active conditions (Fig. 1a).

The findings described above suggest that Hda1C may have two distinct functions in histone deacetylation, namely, deacetylation of histone H3 at inactive genes and deacetylation of histone H4 within active coding regions. To confirm these hypotheses, acetylation patterns were analyzed at two actively transcribing genes: YEF3 and PMA1. Whereas H4 acetylation within the coding regions of these genes was strongly increased in the hda1Δ strain, deletion of HDA1 did not strongly affect H4 acetylation at the promoter regions or H3 acetylation at the promoter or coding regions (Fig. 1b). The same patterns of histone acetylation were also observed in hda2Δ and hda3Δ strains (Supplementary Fig. 1a–c). These findings provide further evidence that Hda1C specifically deacetylates histone H4 within coding regions of actively transcribed genes.

To explore the roles of Hda1C in histone deacetylation on a genome-wide level, we used chromatin immunoprecipitation sequencing (ChIP-seq) to monitor H3 and H4 acetylation in the wild-type and hda1Δ strains. As Hda1C functionally interacts with the general corepressor Tup1, histone acetylation patterns were also determined in tup1Δ cells[15]. Levels of histone acetylation were normalized to total histone H3 levels. In both hda1Δ and tup1Δ cells, H3 acetylation was increased at inactive genes, as sorted by Rpb3 occupancy (Fig. 2a). By contrast, deletion of HDA1 increased H4 acetylation within the coding regions of more active genes (Fig. 2a). We further generated quantitative ChIP-seq data with Schizosaccharomyces pombe spike-in control normalization to ensure the accurate changes in H4 acetylation levels. hda1Δ cells but not tup1Δ cells still exhibited a strong increase of H4 acetylation within more active genes (Fig. 2b). This finding indicates that Hda1C-mediated deacetylation of H4 within coding regions is independent of Tup1.

Next, using the data set from Fig. 2b, we divided the genes into five groups by K-mean clustering to categorize genes based on similarities in acetylation profiles. Approximately 47% of the genes (including groups 3, 4, and 5) showed a considerable increase in H4 acetylation within coding regions upon deletion of HDA1 (Fig. 2c, d). To understand the relationships between the changes in H4 acetylation levels and transcription frequency or gene length, all genes were classified into five groups based on RNA Pol II occupancy in wild-type cells or their length (Fig. 2e, f). Highly active genes in wild-type cells and longer genes showed more substantial increases in H4 acetylation following

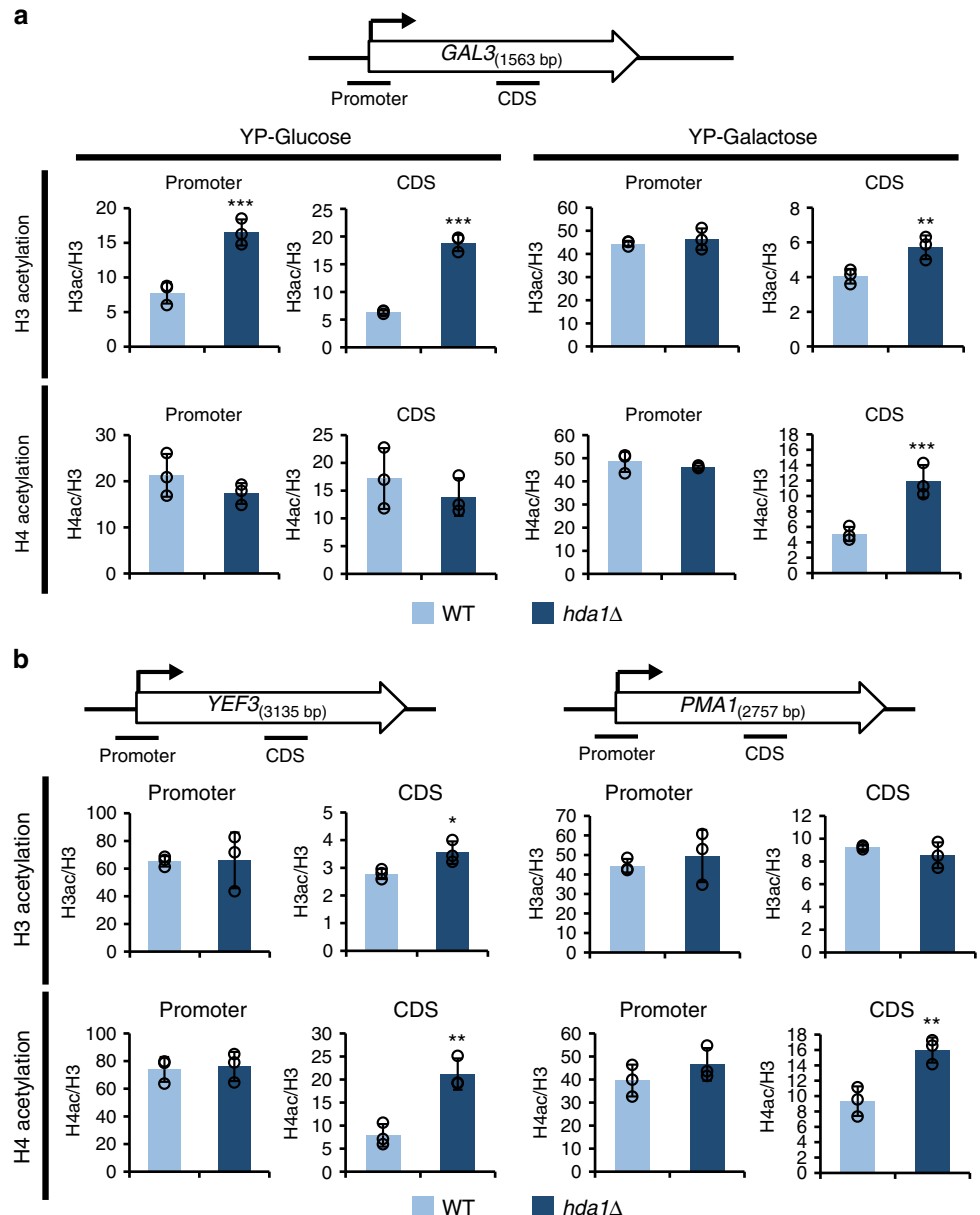

**Fig. 1** Hda1C preferentially deacetylates histone H4 at actively transcribed genes. **a** Hda1C specifically deacetylates histone H4 at active *GAL3*-coding regions. Crosslinked chromatin from the indicated strains grown in YP-Glucose (YPD) or YP-Galactose was precipitated with an anti-H3, anti-acetyl H3, or anti-acetyl H4 antibody as indicated. PCR analysis of the precipitated DNA was carried out on the promoter and coding regions of *GAL3*. A non-transcribed region located close to the telomere of chromosome VI was used as an internal control. The signals for acetyl H3 and acetyl H4 were quantitated and normalized to the total H3 signal. Error bars show the standard deviation calculated from three biological replicates, each with three technical replicates. \*\**p* < 0.01 and \*\*\**p* < 0.001 (two-tailed unpaired Student's *t* tests). **b** Hda1C deacetylates histone H4 at coding regions of the actively transcribed genes *YEF3* and *PMA1*. Crosslinked chromatin from the indicated strains grown in YPD was precipitated with an anti-H3 or anti-acetyl H4 antibody as indicated, and a ChIP assay was performed as in **a**. Error bars show the standard deviation calculated from three biological replicates, each with three technical replicates. \**p* < 0.05 and \*\**p* < 0.01 (two-tailed unpaired Student's *t* tests). Source data are provided as a Source Data file

deletion of *HDA1* than those with lower RNA Pol II occupancies and shorter genes, respectively (Fig. 2e, f). By contrast, deletion of *HDA1* caused a slight increase in H3 acetylation throughout inactive genes (Supplementary Fig. 2a and b). Further analyses of histone acetylation patterns in *tup1Δ* cells revealed that Tup1 primarily controls histone acetylation at inactive promoters (Supplementary Fig. 2c and d). These results suggest that Hda1C preferentially deacetylates histone H4 within coding regions of hyperactive long genes via a Tup1-independent mechanism.

**Hda1 associates with highly transcribed genes**. Although previous studies have proposed that Hda1 targets inactive promoters via Tup1, the observed increase in H4 acetylation within the coding regions of highly transcribed genes following deletion of *HDA1* suggests that it also binds to these regions. Therefore, we performed ChIP experiments using an anti-myc antibody and cells expressing 18-myc-tagged Hda1 (Hda1-myc). Unlike the untagged control, Hda1-myc was highly enriched at the promoter and coding regions of the active genes *YEF3* and *PMA1* (Fig. 3a). By contrast, binding of Hda1-myc was relatively depleted at the

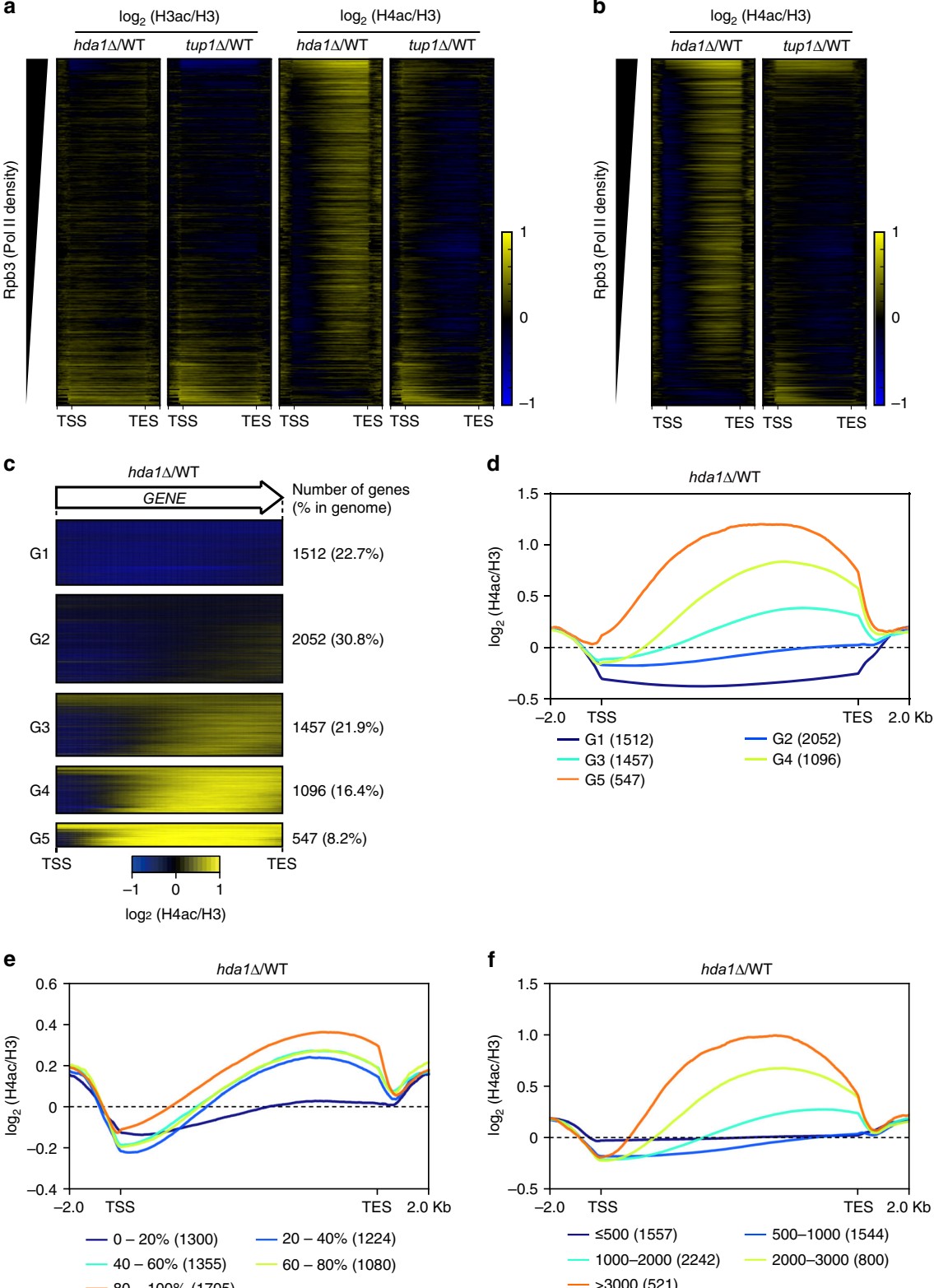

inactive gene *TKL2*, even though Hda1-myc occupancy at this region was approximately threefold higher than that of the untagged control (Fig. 3a).

Next, we examined whether other subunits of Hda1C are required for Hda1 crosslinking. Binding of Hda1 to the *YEF3* and *PMA1* genes was reduced slightly following deletion of *HDA2*,

*HDA3*, or both, even though the Hda1 protein level was similar in wild-type and mutants (Fig. 3b and Supplementary Fig. 3a and b). A previous study showed that Hda2 and Hda3 can bind to double- and single-stranded DNA in vitro[11]. Although binding of these subunits to nascent RNA has not been demonstrated, the strong enrichment of Hda1 at highly active genes suggests that it

**Fig. 2** Actively transcribed long genes are preferentially deacetylated by Hda1C. **a** Hda1, but not Tup1, deacetylates histone H4 within coding regions. Heatmaps of H3 and H4 acetylation levels calculated as the log2 fold change in *hda1Δ* or *tup1Δ* cells versus wild-type cells from two independent ChIP-seq experiments. All genes are sorted by descending order of Rpb3 occupancy from Mayer et al.[45]. The *y* axis indicates each gene and the *x* axis indicates relative position to the transcription start site (TSS) and transcription end site (TES). **b** Confirmation of H4-specific deacetylation by Hda1C. Heatmaps of H4 acetylation levels calculated as in **a** from two independent ChIP-seqs including *S. pombe* chromatin as spike-in controls. All genes are sorted by descending order of Rpb3 occupancy. The *y* axis indicates each gene and the *x* axis indicates relative position to the transcription start site (TSS) and transcription end site (TES). **c** Heatmaps of H4 acetylation patterns for five gene groups identified by K-mean clustering using ChIP-seq data from **b**. The number of genes and % in genome are indicated on the right. **d** Average plot of the data shown in **c**. The *x* axis indicates relative position to the transcription start site (TSS) and transcription end site (TES). The number of genes is indicated in the parenthesis. **e** Actively transcribed genes exhibit a marked increase in H4 acetylation upon deletion of *HDA1*. Average plot of histone acetylation patterns based on RNA Pol II occupancy (Rpb3). The "0–20%" group includes genes showing the lowest Rpb3 occupancy and the "80–100%" group includes genes with the highest level of Rpb3 binding. **f** Longer genes show a larger increase in H4 acetylation than shorter genes upon deletion of *HDA1*. The plot shows average histone acetylation patterns grouped by gene length

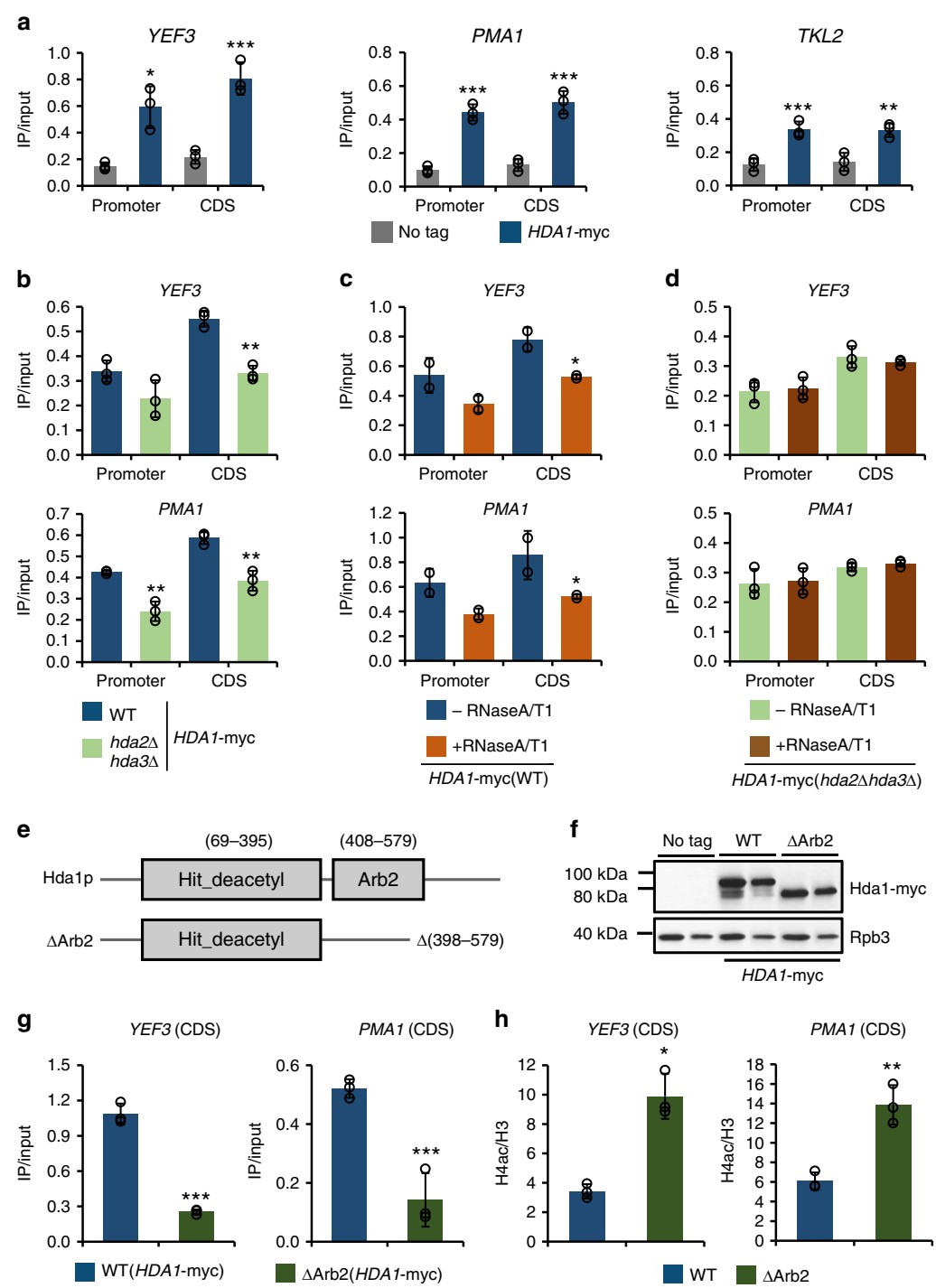

**Fig. 3** The Arb2 domain of Hda1 is required for its binding to chromatin. **a** Hda1 binds strongly to active genes. Crosslinked chromatin from untagged control (*HDA1*) or *HDA1*-myc cells grown in YP-Glucose (YPD) was precipitated with an anti-myc antibody. PCR analysis of the precipitated DNA was carried out on the promoters and coding regions of the indicated genes. A non-transcribed region near the telomere of chromosome VI was used as an internal control. The signals for anti-myc were quantitated and normalized to the input signal. Error bars show the standard deviation calculated from three biological replicates, each with three technical replicates. $*p < 0.05$, $**p < 0.01$, and $***p < 0.001$ (two-tailed unpaired Student's $t$ tests). **b** Hda2 and Hda3 partially affect Hda1 recruitment. ChIP assays using the indicated strains were performed as in **a**. Error bars show the standard deviation calculated from three biological replicates, each with three technical replicates. $**p < 0.01$ (two-tailed unpaired Student's $t$ tests). **c** RNA removal partially reduces Hda1 occupancy. Crosslinked chromatin from cells expressing Hda1-myc was treated with or without RNaseA/T1 and precipitated with an anti-myc antibody. The PCR analysis was performed as in **a**. $*p < 0.05$ (two-tailed unpaired Student's $t$ tests). **d** RNA removal has no effect on Hda1 occupancy in *hda2Δhda3Δ* mutant. Crosslinked chromatin from the indicated strain was treated with or without RNaseA/T1 and precipitated with an anti-myc antibody. The PCR analysis was performed as in **a**. Error bars show the standard deviation calculated from three biological replicates, each with three technical replicates. **e** Schematic representation of the Hda1 protein showing the HDAC domain (Hit_deacetyl; N terminus) and the Arb2 domain (C terminus). The Arb2 domain was deleted at the genomic *HDA1* locus in the ΔArb2 strain. **f** Loss of the Arb2 domain partially reduces Hda1 protein levels. Total extracts from the indicated strains grown in YPD were separated by SDS-PAGE and probed with the indicated antibodies. Rpb3 was used as a loading control. **g** The Arb2 domain is required for Hda1 binding to chromatin. ChIP assay using the indicated strains was done as in **a**. Error bars show the standard deviation calculated from three biological replicates, each with three technical replicates. $***p < 0.001$ (two-tailed unpaired Student's $t$ tests). **h** Loss of the Arb2 domain increases H4 acetylation. ChIP assay using the indicated strains was performed as in Fig. 1b. Error bars show the standard deviation calculated from three biological replicates, each with three technical replicates. $*p < 0.05$ and $**p < 0.01$ (two-tailed unpaired Student's $t$ tests). Source data are provided as a Source Data file

may occur in vivo. To examine whether RNA-binding of Hda1C contributes to its chromatin association, we performed ChIP experiments followed by RNaseA and T1 treatments to remove the RNA. Similar to the effect of *hda2Δhda3Δ*, removal of the RNA reduced the binding of Hda1 to *YEF3* and *PMA1* slightly (Fig. 3c). Importantly, this effect was not seen in *hda2Δhda3Δ* cells (Fig. 3d). These findings suggest that Hda2 and Hda3 contribute partially to Hda1C recruitment, possibly through their RNA-binding activity.

Co-transcriptional histone methylations at H3K4 or H3K36 act as binding sites for other HDACs, including Set3 HDAC and Rpd3S[6–8]. Loss of Set1 or Set2, which are responsible for methylating H3K4 or H3K36, respectively, had no strong effect on Hda1 occupancy at *YEF3* and *PMA1* (Supplementary Fig. 3c). In *set1Δset2Δ* cells, Hda1 binding was reduced only by ~10% suggesting that crosslinking of Hda1 to highly active genes is only partially dependent on co-transcriptional histone methylations at H3K4 and H3K36 (Supplementary Fig. 3c). Ctk1, which phosphorylates serine 2 of the C-terminal domain of Rpb1, was also not required for binding of Hda1 to *YEF3* and *PMA1* (Supplementary Fig. 3d and e).

**Binding of Hda1 to hyperactive genes requires its Arb2 domain**. The Hda1 protein has two distinct domains: a HDAC domain at its N-terminal region and an Arb2 domain at its C terminus (Fig. 3e). The recombinant Arb2 domain is known to interact strongly with nucleosomes in vitro, even though it has no role in histone deacetylation by Hda1 in vitro[11,12]. Deletion of the Arb2 domain reduced Hda1 protein levels by ~13% *in vivo* ($p = 0.002943$) ($n = 3$) (Fig. 3f). Surprisingly, loss of this domain led to a dramatic decrease in binding of Hda1 to the active genes *YEF3* and *PMA1* (Fig. 3g). Consistently, the H4 acetylation pattern of the cells expressing Hda1 lacking the Arb2 domain was comparable to that of the *hda1Δ* cells (Fig. 3h). These results suggest that the Arb2 domain is important for Hda1C binding to highly active genes.

Next, we used quantitative ChIP-seq experiments using normalizing *S. pombe* spike-in controls to confirm Hda1C crosslinking to highly active genes and the role of the Arb2 domain in this process. Hda1 was highly enriched at actively transcribed genes and this binding was observed throughout all yeast chromosomes (Fig. 4a and Supplementary Fig. 4a). This occupancy was reduced markedly upon deletion of the Arb2

domain (Fig. 4a). The top 25% of genes with the highest level of RNA Pol II also showed the Arb2 domain-dependent association of Hda1 with coding regions (Fig. 4b). In addition, this binding showed a weak, but statistically significant correlation with increases of H4 acetylation in *hda1Δ* cells (Fig. 4c and Supplementary Fig. 4b). Furthermore, group 4 and 5 in Fig. 2d having the highest level of H4 acetylation upon *HDA1* deletion showed relatively high levels of RNA Pol II (Supplementary Fig. 4c). For example, binding of Hda1 to two highly active genes, *TEF1* and *TEF2* was correlated with Rpb3 occupancy and this binding was greatly reduced upon deletion of the Arb2 domain (Fig. 4d). In addition, H4 acetylation, but not H3 acetylation, at these genes was increased in *hda1Δ* cells (Fig. 4d). In contrast, only H3 acetylation was increased at an inactive gene, *TKL2* (Fig. 4d). Collectively, these data indicate that Hda1 binds directly to hyperactive genes, possibly via its Arb2 domain, to specifically deacetylate histone H4.

An obvious question is whether binding of Hda1 to highly active genes is transcription-dependent. To examine this possibility, the interaction between Hda1 and Rpb3, a subunit of RNA Pol II, was examined by co-immunoprecipitation assay. Paf1, a component of the PAF complex, interacted weakly with Rpb3 (Fig. 4e). However, a strong interaction between Hda1 and Rpb3 was observed (Fig. 4e). Importantly, this interaction was almost absent when the Arb2 domain is deleted (Fig. 4f). These results suggest that Hda1C could be recruited to highly transcribed regions via the interaction between the Arb2 domain of Hda1 and RNA Pol II (Fig. 4f).

To test this possibility, we determined the occupancies of Hda1-myc and RNA Pol II (Rpb3) at various positions of the *GAL1* and *GAL3* genes before and after glucose shut-off. Rpb3 strongly bound to the *GAL* genes in medium containing galactose where *GAL* genes are active. This binding was completely abolished at 4 min post transfer to glucose medium (Fig. 4g). Similarly, Hda1 binding to these genes was also high in galactose medium, but Hda1 rapidly dissociated following transfer to glucose medium, indicating that RNA Pol II transcription is critical for targeting of Hda1C to highly active genes (Fig. 4g). Consistent with this finding, a previous study also showed transcription-dependent binding of Hda1 to *ARG1* gene[16]. Histone acetylation patterns were also examined to determine whether Hda1C binding is directly associated with histone deacetylation. During growth in galactose medium, H4

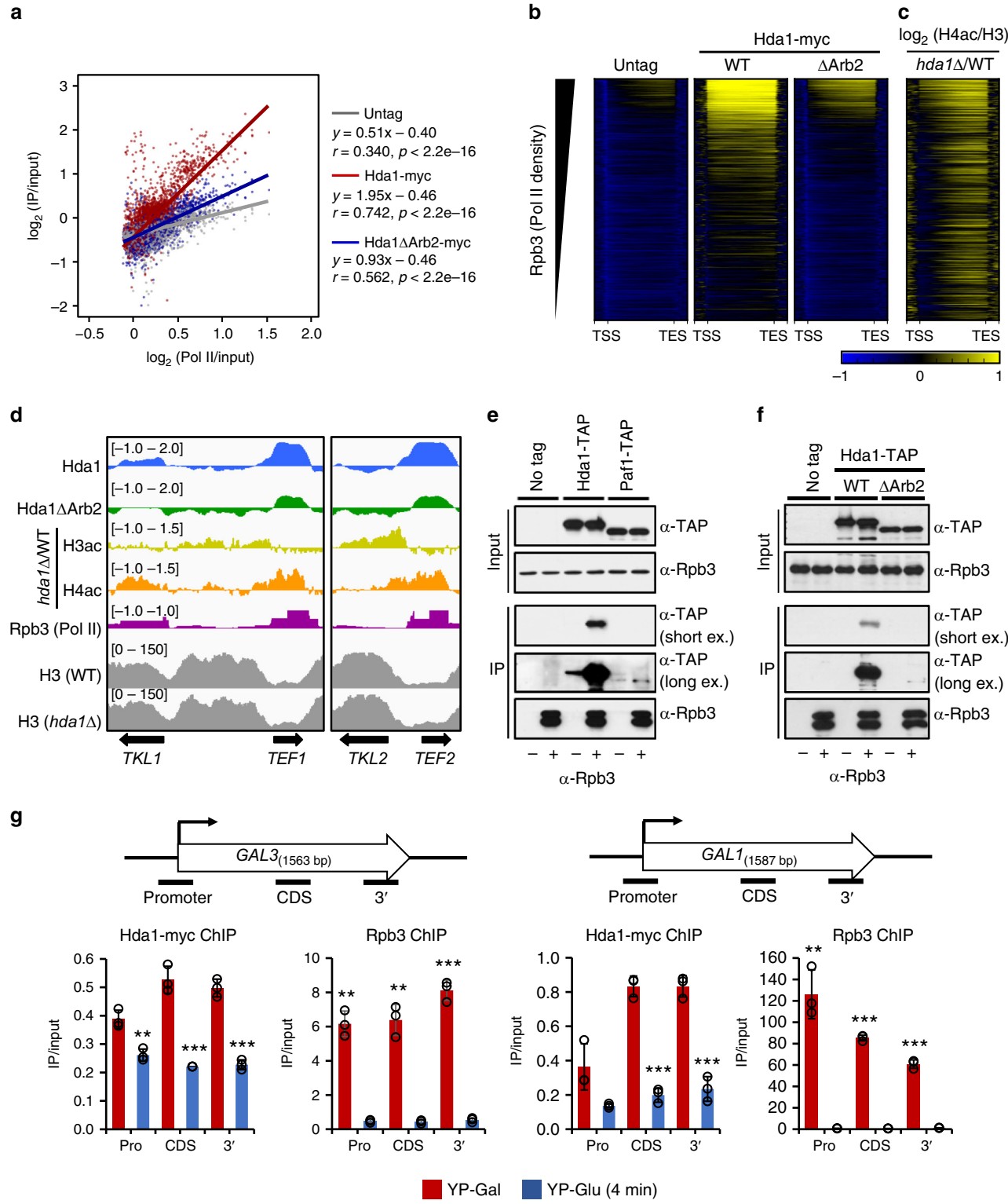

acetylation was increased within the coding region but not the promoter of *GAL3* in the absence of Hda1 (Fig. 1a and Supplementary Fig. 4d). However, this effect of Hda1 depletion was not seen at 4 min after glucose shut-off. Instead, H3 acetylation was increased at this time-point in *hda1Δ* cells (Supplementary Fig. 4d), suggesting a dynamic switch of the specificity of Hda1C from histone H4 to H3 depending on the transcription status. Overall, these findings indicate that transcription-dependent targeting of Hda1C via the interaction between the Arb2 domain of Hda1 and RNA Pol II facilitates histone H4-specific deacetylation at highly active genes.

**Differential deacetylation of histone H4 by Hda1C and Set2.** Co-transcriptional histone methylation at H3K36 by Set2 recruits and/or activates the Rpd3S HDAC to reduce histone acetylation in the 3′ regions of genes[7,8]. A previous study proposed that the Set2–Rpd3S pathway acts primarily on infrequently transcribed

**Fig. 4** Transcription-dependent binding of Hda1C to active coding regions. **a** Hda1 binds to hyperactive genes. Scatterplot of Hda1 occupancy from two independent ChIP-seqs including *S. pombe* chromatin as spike-in controls in untagged (gray), Hda1-myc wild-type (red), and Hda1-myc (ΔArb2) (blue) cells, plotted against Rpb3 levels (relative Pol II). Pearson's correlation coefficients are indicated. **b** Hda1 occupancies for the top 25% of highly transcribed genes (1705 genes) from two independent ChIP-seqs including *S. pombe* chromatin as spike-in control. The genes were sorted in descending order of Rpb3 occupancy in wild-type cells. The *y* axis indicates each gene and the *x* axis indicates relative position to the transcription start site (TSS) and transcription end site (TES). **c** Histone acetylation patterns of the genes shown in **b**. **d** ChIP-seq tracks at active genes, *TKL1*, *TEF1*, and *TEF2*, and an inactive gene, *TKL2* showing the signals for Hda1 occupancy, H4 acetylation levels, H3 acetylation levels, Rpb3 occupancy, or histone H3 occupancy. **e** Hda1 interacts with Rpb3. Total extracts prepared from the indicated strains were incubated with an anti-Rpb3 antibody and protein G beads. The precipitates (IP) were analyzed by immunoblotting for TAP-tagged proteins (TAP) and Rpb3. Two independent experiments showed the same results. **f** The Arb2 domain is important for the interaction between Hda1 and Rpb3. Co-immunoprecipitation assay using the indicated strains was done as in **e**. Two independent experiments showed the same results. **g** Transcription-dependent Hda1C binding to *GAL* genes. The cells were grown in YP-Galactose (red) and then shifted to YP-Glucose for 4 min (blue). ChIP analyses of *GAL3* and *GAL1* were performed using anti-Rpb3 or anti-myc antibodies, as in Fig. 3a. Error bars show the standard deviation calculated from three biological replicates, each with three technical replicates. **p < 0.01 and ***p < 0.001 (two-tailed unpaired Student's *t* tests). Source data are provided as a Source Data file

long genes[17]. This function contrasts with that of Hda1C, which appears to preferentially deacetylate histone H4 within the coding regions of hyperactive genes (Fig. 2a). To confirm the distinct functions of Hda1C and the Set2–Rpd3S pathway, we compared the H4 acetylation patterns of *hda1Δ* and *set2Δ* cells. As reported previously, deletion of *SET2* mainly increased H4 acetylation within less active genes, whereas increased H4 acetylation in *hda1Δ* cells was correlated with transcription frequency (Fig. 5a, b). As H4 acetylation was increased at many genes in the *hda1Δ* and *set2Δ* strains, we examined whether Set2 and Hda1C functionally interact. First, we performed peptide pull-down experiments to monitor the interaction between Hda1C and trimethylated H3K36 (H3K36me3). NuA3 HAT is known to have at least two subunits, Nto1 and Pdp3, that directly recognize H3K36me3[18,19]. Although a strong interaction between Nto1 and H3K36me3 was seen, Hda1C did not recognize H3K36me3 even in low salt conditions (Supplementary Fig. 5a and b). In addition, loss of Hda1 had no effect on H3K36me3 levels in the *YEF3*- and *PMA1*-coding regions (Supplementary Fig. 5c). H4 acetylation at the coding regions of two genes, *STT4* and *HAP1*, was significantly increased in mutants for Hda1 and Set2. A double deletion strain lacking Hda1 and Set2 showed higher levels of H4 acetylation than the single deletion mutants (Fig. 5c). Taken together, these findings suggest that the Hda1C and the Set2–Rpd3S pathway differentially control histone acetylation within coding regions.

To understand the distinct functions of Hda1C and Set2 further, genes displaying hyperacetylation of H4 within coding regions following *HDA1* or *SET2* deletion were divided into three groups: Hda1-specific, Set2-specific, and common genes. Approximately 68% of the genes displayed H4 hyperacetylation in both mutants (1863 of 2739 and 2734 genes in *hda1Δ* and *set2Δ* cells, respectively, corresponding to 68.0% and 68.1%, respectively; Fig. 5d–g, yellow). The proportions of genes that were Hda1-specific (876 of 2739 genes, 32.0%; Fig. 5d–g, green) and Set2-specific (871 of 2734 genes, 31.9%; Fig. 5d–g, red) were comparable. Surprisingly, the Set2-specific genes showed lower levels of H3K36me3 than the Hda1-specific genes (Fig. 5d, e), suggesting that Set2 may affect histone acetylation independently of H3K36 methylation or Rpd3S at these genes. Gene Ontology analyses of the Hda1-specific, Set2-specific, and common genes revealed their distinct biological functions (Supplementary Fig. 5d). Whereas Hda1-specific genes were linked to cytoplasmic translation and biosynthetic processes, Set2-specific genes were enriched for adaptation of signaling pathways and base-excision repair (Supplementary Fig. 5d). Overall, these results indicate that Hda1C functions independently of the Set2–Rpd3S pathway to deacetylate histone H4 at highly transcribed genes.

**Hda1C may negatively affect elongation**. Histone acetylation/deacetylation within transcribed regions affects RNA Pol II elongation[3]. The Set2–Rpd3S pathway negatively regulates elongation. Cells that are defective in this pathway show resistance to 6-azauracil or mycophenolic acid and bypass the requirement for Bur1, a positive elongation factor[8]. In addition, the Set2–Rpd3S pathway inhibits initiation from cryptic promoters and suppresses histone exchange[7,20].

Interestingly, loss of Hda1C slightly suppressed the growth defect of *bur1Δ* (Fig. 5h). A previous study showed that overexpression of the histone demethylases Jhd1 and Rph1, which target H3K36 methylation, partially supports the growth of *bur1Δ* cells[21]. A synergistic effect on cell growth was seen when Jhd1 or Rph1 was overexpressed in *hda1Δ* cells, suggesting that Hda1C may inhibit RNA Pol II elongation via H4-sepcific deacetylation at highly transcribed genes (Supplementary Fig. 5e). Although the Set2–Rpd3S pathway represses cryptic promoters within coding regions by deacetylating histones, deletion of *HDA1* had no effect on cryptic initiation from the *PCA1* and *STE11* genes (Supplementary Fig. 5f and g). Unexpectedly, genes with H4 hyperacetylation in *hda1Δ* cells had relatively high histone occupancy (Supplementary Fig. 5h). Furthermore, longer genes also showed a slightly increased histone density within coding regions (Supplementary Fig. 5i). In contrast, reduced histone density was seen at 3′-transcribed regions in *set2Δ* cells (Supplementary Fig. 5h and i). Although how Hda1C maintains optimal histone occupancy remains elusive, the absence of cryptic initiation in mutants for Hda1C may be due to increased histone density. Taken together, these results suggest that Hda1C-mediated H4 deacetylation at highly active genes may negatively regulate RNA Pol II elongation.

**H3 deacetylation by Hda1C delays gene induction**. Although chromatin regulators have been thought to play important roles in global gene regulation, little or no obvious effect on global gene expression was seen in a steady-state condition[22]. Instead, they seem to function as modulators that regulate the kinetics of transcriptional responses to support cell adaptation and fitness upon environmental changes[3,23].

Hda1C and Tup1 contribute to gene repression and both proteins were required for H3 deacetylation at inactive genes in our ChIP-seq analyses (Fig. 2a). Previous studies showed that Hda1C had a minimal effect on global transcription under steady-state conditions but affected the kinetics of gene induction upon diamide stress[22,23]. To further understand the exact function of Hda1C, we monitored the transcriptional responses of wild-type and *hda1Δ* cells during carbon source shifts[24,25]. In this experiment, wild-type, *hda1Δ*, and *hda3Δ* cells were initially

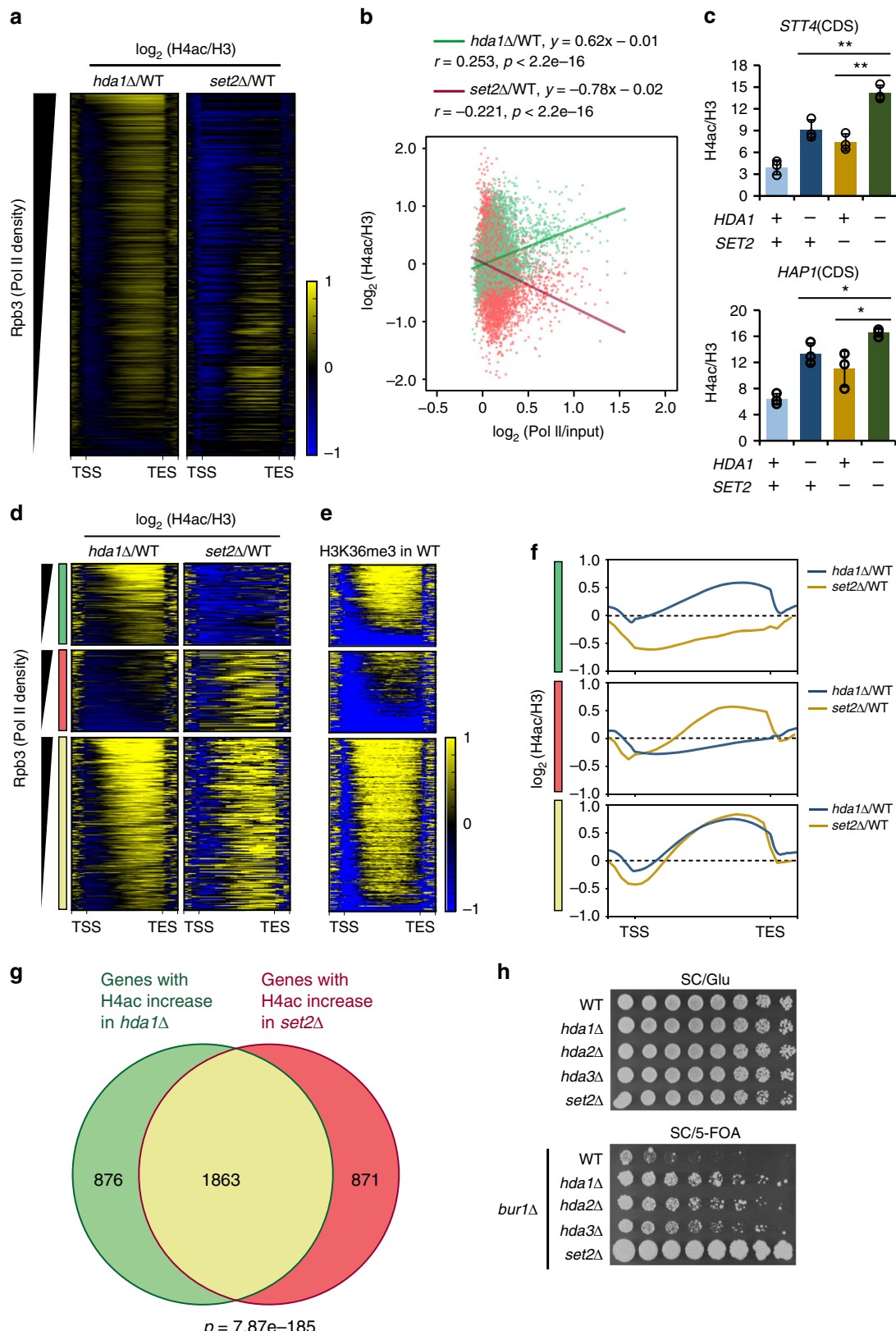

grown in synthetic complete medium containing raffinose, and subsequently shifted to galactose for 120 min, glucose for 120 min, and then back to galactose for 30 min (Fig. 6a). The basal expression level of *HXT5*, a gene induced by galactose, was not affected by deletion of *HDA1* or *HDA3* in raffinose medium;

however, a strong and rapid induction was seen in the Hda1C mutants during galactose incubation (Supplementary Fig. 6a).

To explore how Hda1C affects global gene expression dynamics, total RNAs from the wild-type and *hda1Δ* cells were analyzed by strand-specific RNA-sequencing. Our previous

**Fig. 5** Hda1C and Set2 differentially affect H4 acetylation within coding regions. **a** Heatmaps of H4 acetylation patterns in *hda1Δ* (6664 genes) and *set2Δ* (5648 genes) cells. The plots show the log2 fold changes in H4 acetylation in *hda1Δ* (from Fig. 2b) or *set2Δ* cells versus wild-type cells. All genes are sorted by descending order of Rpb3 occupancy. The y axis indicates each gene and the x axis indicates relative position to the transcription start site (TSS) and transcription end site (TES). Histone acetylation patterns of *set2Δ* were analyzed using the data set from Venkatesh et al.[28]. **b** H4 deacetylation by Hda1, but not Set2, correlates with RNA Pol II occupancy. Scatterplot showing the relationship between H4 acetylation changes in *hda1Δ* or *set2Δ* cells and Rpb3 occupancy. Pearson's correlation coefficients are indicated. **c** Dual loss of Hda1 and Set2 increases H4 acetylation further. A ChIP assay using the indicated strains was performed as in Fig. 1b. *$p < 0.05$ and **$p < 0.01$ (two-tailed unpaired Student's $t$ tests). Source data are provided as a Source Data file. **d** Genes displaying H4 hyperacetylation in the *hda1Δ* or *set2Δ* cells were divided into three groups: Hda1-specific (green), Set2-specific (red), and common (yellow) genes. Heatmaps of H4 acetylation patterns are shown as in **a**. The y axis indicates each gene and the x axis indicates relative position to the transcription start site (TSS) and transcription end site (TES). **e** Heatmap of H3K36me3 patterns in the three groups shown in **d**. The y axis indicates each gene and the x axis indicates relative position to the transcription start site (TSS) and transcription end site (TES). H3K36me3 patterns were analyzed using the data set from Weiner et al.[46]. **f** Average plots of H4 acetylation patterns in the three groups shown in **d**. **g** Venn diagram of the three groups described in **d**. The significance of the overlap was calculated using Fisher's exact test. **h** Loss of Hda1C bypasses the requirement of Bur1, a positive elongation factor. *BUR1* plasmid shuffling strains harboring the indicated deletions were spotted in threefold dilutions onto a synthetic complete (SC)/Glu plate (shown after 2 days) or a SC/5-FOA plate (shown after 8 days). The *set2Δ* mutant was used as a positive control

studies identified ~1000 genes that differentially respond to distinct carbon sources[24,25]. Hda1C-regulated genes were identified as those showing at least a 1.7-fold increase in transcript levels at one or more time-points during carbon source shifts. A total of 331 genes induced during galactose incubation were negatively regulated by Hda1C (Fig. 6b). The transcript levels of these genes were similar in *hda1Δ* and wild-type cells during incubation in raffinose medium, but deletion of *HDA1* resulted in rapid and strong induction during galactose incubation (Fig. 6b). Notably, deletion of *HDA1* had no strong effect on the induction of these genes when they were hyperactivated by the second galactose pulse (Fig. 6b).

Next, the histone acetylation patterns of the 331 Hda1C-regulated inducible genes were analyzed. Although Hda1C functionally interacts with Tup1, the histone acetylation patterns differed slightly in *hda1Δ* and *tup1Δ* cells. In *hda1Δ* cells, H3 acetylation, but not H4 acetylation, was increased at the promoter regions (Fig. 6c) and within gene bodies (Fig. 6d). However, deletion of *TUP1* increased the acetylation of both H3 and H4 at promoter regions, suggesting that Tup1 might functionally interact with Rpd3 to promote H4-specific deacetylation at inactive promoters[26] (Fig. 6c, d). These results indicate that, in addition to H4-specific deacetylation within hyperactive coding regions, Hda1C preferentially deacetylates H3 to delay the kinetics of gene induction upon environmental changes.

## Discussion

Histone acetylation and deacetylation within coding regions play important roles in RNA Pol II elongation[27]. The Set2–Rpd3S HDAC pathway slows elongation and inhibits transcription initiation from cryptic promoters[7,8]. In addition, H3K36 methylation by Set2 inhibits the incorporation of acetylated histones to suppress histone exchange within coding regions[28]. The Set3 HDAC also represses initiation from internal promoters in 5′-transcribed regions[6]. The results presented here indicate that Hda1C specifically deacetylates histone H4, but not H3, within hyperactive genes to partially inhibit elongation. In addition, Hda1C preferentially deacetylates histone H3 at inactive genes to delay the kinetics of gene induction upon environmental changes (Fig. 7).

Although Hda1C binds to highly active genes, it lacks domains recognizing co-transcriptional histone modifications, including H3K36me3 and H3K4me2, enriched in transcribed regions. Instead, the Arb2 domain of Hda1 and the N-terminal regions of Hda2 and Hda3 contribute to targeting of the complex to chromatin. Recruitment of Hda1C to hyperactive genes is likely initiated by RNA Pol II transcription. Hda1C interacts directly

with Rpb3 via the Arb2 domain of Hda1 and its crosslinking is highly correlated with transcription frequency (Fig. 4a, b). Furthermore, our ChIP data indicated that Hda1C travels with RNA Pol II at *GAL* genes (Fig. 4g). Therefore, we propose that RNA Pol II initially recruits Hda1C to highly transcribed genes, and then multiple interactions including Arb2-RNA Pol II, Arb2-histones, and DNA/RNA-Hda2 and -Hda3 stabilize Hda1C binding to highly active genes. Notably, some mammalian HDACs also bind active genes[29]. In particular, HDAC6 is highly enriched at active coding regions via an interaction with the phosphorylated C-terminal domain of RNA Pol II, although its function in histone deacetylation remains unknown.

Although Hda1C and the Set2–Rpd3S pathway play important roles in histone deacetylation, they differentially regulate H4 acetylation within coding regions (Fig. 7). Whereas Hda1C is more active at highly transcribed genes, the Set2–Rpd3S pathway preferentially acts on infrequently transcribed genes (Fig. 5a, b). Furthermore, mutants for the Set2–Rpd3S pathway, but not Hda1C mutant cells, showed reduced H4 acetylation at transcription start sites (Fig. 5d, f). These findings suggest independent roles of these HDACs in histone deacetylation. Genes transcribing at intermediate levels are deacetylated by both Hda1C or the Set2–Rpd3S pathway, but these HDACs may act at different times. It is possible that Hda1C deacetylates histone H4 at currently transcribing genes via an interaction with RNA Pol II, whereas the Set2–Rpd3S pathway deacetylates histone H4 at the same genes when they are not currently transcribing by recognizing H3K36me3. Hda1C may deacetylate histone H4 immediately after RNA Pol II passes, but acetylated histone H4 incorporated during nucleosome reassembly may be deacetylated by the Set2–Rpd3S pathway (Fig. 7).

Loss of the Set2–Rpd3S pathway leads to an accumulation of cryptic and antisense transcripts from internal promoters within coding regions[7,17,30]. As expected, the *STE11* and *PCA1* genes produced cryptic transcripts in *set2Δ* cells (Supplementary Fig. 5f and g). Some Set2-repressed cryptic transcripts, including *PCA1*, are transcriptionally induced upon environmental changes[25]. Although deletion of *HDA1* slightly increased H4 acetylation at *PCA1* but not at *STE11*, cryptic transcripts were not observed, suggesting that elevated H4 acetylation itself is not sufficient to activate cryptic promoters (Supplementary Fig. 5f and g). Unexpectedly, loss of Hda1 results in a slight increase in histone occupancy. Although how Hda1C reduces nucleosome occupancy remains unclear, a slight increase of histone levels may block initiation from cryptic promoter in *HDA1* deleting cells (Supplementary Fig. 5h and i). Cells defective in the Set2–Rpd3S pathway bypassed the requirement for Bur1, a positive elongation factor[8] (Fig. 5h). The growth defect of *bur1Δ* cells was also

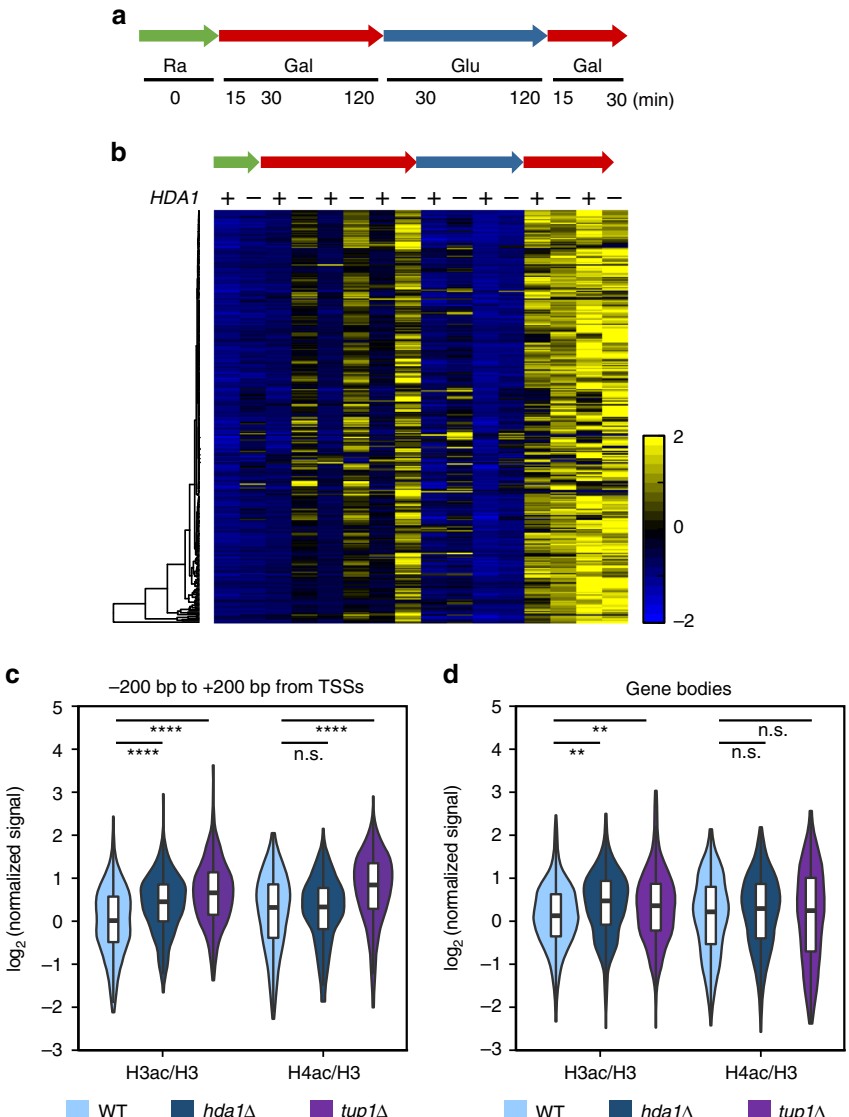

**Fig. 6** H3 deacetylation by Hda1C fine-tunes the kinetics of gene induction. **a** Schematic representation of the time-course experiments to monitor gene expression changes in wild-type and *hda1Δ* cells during carbon source shifts. Ra, raffinose; Gal, galactose; Glu, glucose. RNA samples were collected at the indicated time-points. **b** Hda1C negatively regulates the kinetics of gene induction. RNA samples from the time-course experiments described in **a** were analyzed by strand-specific RNA-sequencing. The ratios of transcript levels for 331 genes in *hda1Δ* versus wild-type cells are shown. Hda1-repressed genes were identified as those showing at least a 1.7-fold increase in transcript levels at one or more time-points. **c**, **d** Hda1C preferentially deacetylates histone H3 at inducible genes. Average profiles of H3 and H4 acetylation changes upon deletion of *HDA1* or *TUP1* around the transcription start sites (TSSs; −200 bp to +200 bp) (**c**) and within gene bodies (**d**) of the Hda1-repressed genes. Significance levels were computed by permutation tests. \*\**p* < 0.01 and \*\*\*\**p* < 1.0 × 10$^{-10}$

suppressed by overexpression of the histone demethylases Jhd1 and Rph1, which demethylate H3K36 methylation[21] (Supplementary Fig. 5e). Notably, deletion of *HDA1* also partially supported the growth of *bur1Δ* cells and a synergistic effect was seen when Jhd1 or Rph1 was overexpressed (Fig. 5h and Supplementary Fig. 5e). Taken together, these findings suggest that Hda1C-mediated deacetylation of histone H4 within coding regions may inhibit RNA Pol II elongation.

Although loss of Hda1 resulted in a significant increase in histone acetylation throughout the yeast genome, there was little effect on global transcript levels under a steady-state condition[22]. Recent studies have suggested that chromatin regulators may fine-tune the kinetics of transcriptional responses upon environmental changes[3]. Here, we identified 331 genes that were negatively regulated by Hda1C. Whereas basal transcript levels of

these genes were not strongly affected by deletion of *HDA1*, their induction by switching to galactose medium was stronger and more rapid in *hda1Δ* cells than in wild-type cells (Fig. 6b). Loss of Hda1 increased H3 acetylation but not H4 acetylation at these genes, suggesting that Hda1C-mediated deacetylation of H3 at inactive genes delays gene induction (Fig. 6c, d).

An important question is how Hda1C differentially deacetylates histone H3 or H4 at a different or the same locus, depending on the transcriptional status of the gene. Rpd3-containing HDACs can deacetylate histones H3 and H4 at the same locus[6]. Furthermore, the Set3 HDAC includes two HDACs, Hst1 and Hos2, which preferentially deacetylate histones H3 and H4, respectively[6]. Previous reports and our data suggest that, although Hda1C can deacetylate histones H3 and H4, its specificity is tightly regulated depending on the transcription frequency.

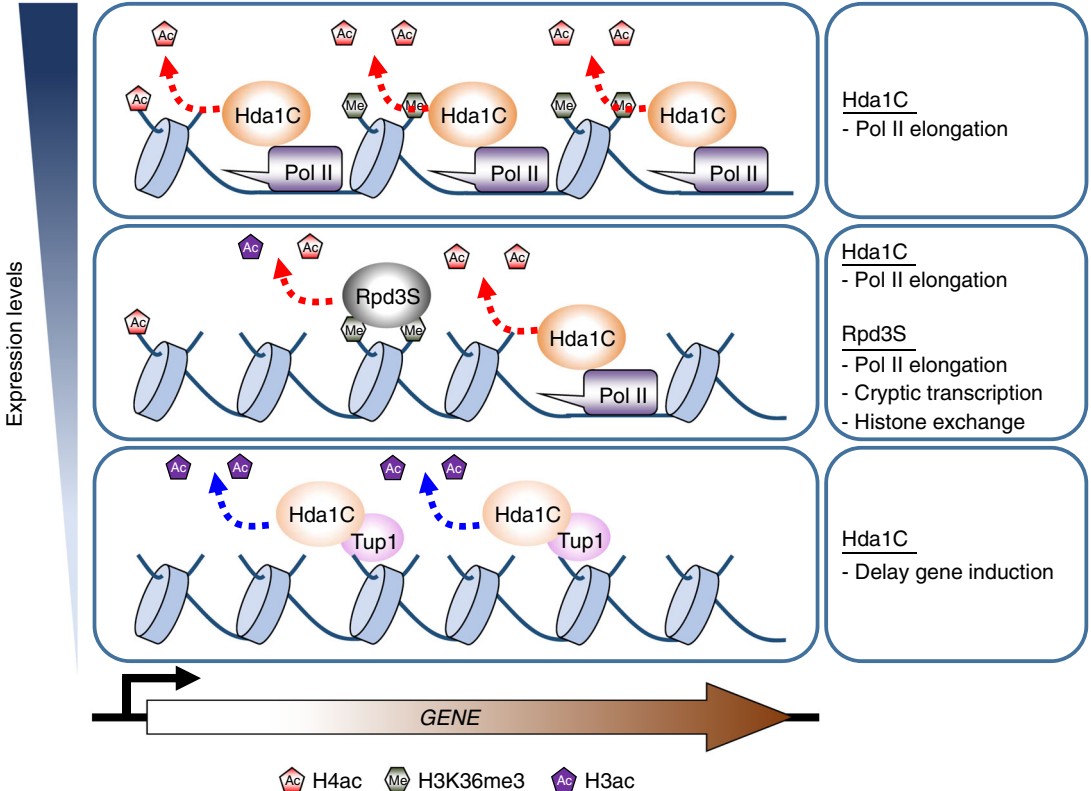

**Fig. 7** Model for the regulation of histone acetylation by Hda1C. At hyperactive genes, Hda1C strongly interacts with RNA Pol II to specifically deacetylate histone H4 within coding regions. Deacetylation of H4 by Hda1C may inhibit RNA Pol II elongation. At genes transcribing at intermediate levels, both Hda1C and Rpd3S promote histone deacetylation. When a gene is currently transcribing, Hda1C interacting with RNA Pol II may deacetylate histone H4. When a gene is not being transcribed, or once RNA Pol II passes, recognition of H3K36me3 by the Eaf3 chromodomain of Rpd3S may result in deacetylation of the remaining acetylated histones. Both Hda1C and Rpd3S may negatively affect elongation. In addition, Rpd3S inhibits initiation from cryptic promoters and suppresses histone exchange. At inactive or inducible genes, Hda1C preferentially deacetylates histone H3 to delay the kinetics of gene induction

Whereas Hda1C mainly reduces H3 acetylation at transcriptionally inactive genes, histone H4 is specifically deacetylated by this complex at highly transcribed genes (Fig. 7). Post-translational modifications of Hda1C might affect its specificity. Hda1 and Hda3 are highly sumoylated upon KCl or sorbitol stress[31]. In addition, these two proteins are phosphorylated by unknown kinases[32]. Determining whether these modifications affect the specificity of Hda1C will be important to understand the exact function of Hda1C in histone deacetylation and transcription regulation.

## Methods

**Yeast strains**. Yeast strains used in this study are listed in Supplementary Table 1. The time-course experiments were done with matched *HDA1* and *hda1Δ* strains as previously described[24]. To generate the mutant lacking the Arb2 domain in Fig. 3, the *delitto perfetto* strategy was used[33]. The *HDA1*-myc strain was generated by inserting the 18-myc tag into C-terminus of *HDA1*. Except for Figs. 1a, 4g and Supplementary Fig. 4d, all strains were grown in 200 ml of YPD (Yeast extract Peptone Dextrose; 1% Yeast Extract, 2% Peptone, 2% Dextrose (Glucose)) at 30℃ until OD600 was 0.5–0.6. For Fig. 1a, the indicated strains were grown in YPD or YP-Galactose (1% Yeast Extract, 2% Peptone, 2% Galactose). For Fig. 4g, and Supplementary Fig. 4d, cells were grown in YP-Galactose media to an optical density at 600 nm of 0.6 and subsequently transferred to YPD for 4 min.

**RT-PCR**. RNA was extracted from cells with hot phenol. Total RNAs was treated with DNase I (Thermo Fisher Scientific) and first-strand cDNA was prepared with 1 μg total RNA, ReverTra Ace qPCR RT kit (TOYOBO), and gene-specific primers. cDNA was analyzed by real-time qPCR using SYBR Green Supermix and CFX96 cycler (Bio-Rad).

The sequences of oligonucleotides used in this study are listed in Supplementary Table 2.

**Peptide pull-down assay**. Whole cell extracts were prepared with binding buffer (50 mM Tris-HCl (pH 7.5), 0.1% NP-40) containing 150 mM, 200 mM, or 300 mM NaCl and protease inhibitors. In all, 1 mg of total extracts was incubated with 1μg of biotinylated histone peptides (Anaspec) and 25 μl of streptavidin coupled Dynabeads (Invitrogen) at 4℃ overnight. Beads were washed five times with 1.5 ml of binding buffer and precipitates were resolved by sodium dodecyl sulfate polyacrylamide gel electrophoresis (SDS-PAGE) followed by immunoblot analysis.

**Northern blot analysis**. Total RNA was isolated from cells with hot phenol and 10 μg of total RNA was separated on an agarose gel and then transferred to nylon membrane. Northern blot analysis was done as previously described[34]. The sequences of oligonucleotides used for northern blot analysis are listed in Supplementary Table 2. Strand-specific probes were generated by unidirectional PCR in the presence of [α-32P] dATP with only one primer. Hybridization was done in a buffer containing 1% BSA, 7 % SDS, 1 mM EDTA (pH 8.0), and 300 mM Sodium phosphate buffer (pH 7.2). The membranes were washed with 2× SSC and 0.1% SDS for 20 min and exposed to PhosphoImager. Uncropped Northern-blots can be found in Supplementary Fig. 9.

**Spot assay**. For spotting analysis, cells were resuspended at $2.5 \times 10^8$/ml and subjected to threefold serial dilutions in synthetic complete (SC) media lacking any carbon sources, and 3 μl of aliquot of each dilution was spotted on the indicated plates. Growth was assayed at 2 days (SC/Glu as a positive control) and 8 days (SC/5-FOA to select against the *BUR1/URA3* plasmid).

**Western blot analysis**. Cells expressing TAP-tagged or myc-tagged proteins were grown in YPD at 30 ℃ to mid-log phase. Cells were lysed using lysis buffer (50 mM Tris, pH 7.5, 150 mM NaCl, 0.1% NP-40) with protease inhibitors (Pepstatin A 1 μM, Aprotinine 0.3 μM, Leupeptin 1 μM, phenylmethylsulfonyl fluoride 1 mM) and

glass beads. Protein concentration was quantitated by Bradford assay. For SDS-PAGE and western blot analyses, 15~30 μg of whole cell extracts was used. Proteins were separated in SDS-PAGE and transferred onto a PVDF membrane (Millipore). The blots were visualized on film with SuperSignal West Pico Chemiluminescent Substrate (Thermo Fisher Scientific). Uncropped Western-blots can be found in Supplementary Figs. 7 and 8.

**Co-immunoprecipitation assay.** Whole cell extracts were prepared with binding buffer (50 mM Tris-HCl (pH 7.5), 0.1% NP-40) containing 150 mM NaCl and protease inhibitors (1 μM aprotinin, 1 μM leupeptin, 1 μM pepstatin, 1 mM phenylmethylsulfonyl fluoride). In all, 1.5 mg of total extracts was incubated with 25 μl of protein G sepharose (GE Healthcare) and with or without 4 μl of α-Rpb3 antibody (BioLegend) at 4 °C for 4 h. Beads were washed five times with 1.5 ml of binding buffer and precipitates were resolved by SDS-PAGE followed by immunoblot analysis.

**Chromatin immunoprecipitation and ChIP-sequencing.** Cells fixed with 1% formaldehyde were subjected to ChIP as previously described[6] with oligonucleotides as listed in Supplementary Table 2. The following histone antibodies were used: anti-H3K36me3 (Abcam ab9050), anti-acetyl H4 (Millipore 06–598), anti-acetyl H3 (Millipore 06–599), anti-myc (BioLegend 626802) and anti-H3 (Abcam ab1791). Except for H3 acetylation, all ChIP-seqs included *S. pombe* "spike-in" added at 10% relative to *Saccharomyces cerevisiae* chromatin.

Precipitated DNAs were analyzed by quantitative real-time PCR using THUNDERBIRD® SYPR qPCR Mix (TOYOBO) and CFX96 cycler (Bio-Rad). The DNA libraries for ChIP-seq were prepared using Accel-NGS 2 S Plus DNA Library Kit (Swift Biosciences) and sequenced on the HiSeq2500 platform (Illumina) following the manufacturer's instructions.

**Sequence analysis.** To ensure an accurate comparison between experimental conditions, genomic DNA fragments of *S. pombe* were used as a spike-in control. The sequence reads were aligned to the *S. cerevisiae* genome (R61–1–1) or *S. pombe* genome (ASM294v2) using Bowtie2 program (ver. 2.2.5)[35] and ones exclusively assigned to each genome were counted to remove ambiguity. Data filtering, conversion, and visualization were performed by Picard-tools-2.6 (http://broadinstitute.github.io/picard), BEDTools-2.26.0[36] and IGV-2.3.91, respectively[37]. Assuming that spike-ins should contain the constant amount of DNA in each experiment, the total sequence read numbers were adjusted and the ratio of the read numbers from treatment (H4ac or H3ac) to the ones from control (H3) was calculated as the enrichment factor in *S. pombe*. Considering this ratio, the relative enrichment of modified histones (H4ac or H3ac) to the control (H3) in *S. cerevisiae* was calculated by MACS2 (ver. 2.1.0) program with parameter '–nofilter'[38] and deepTools-2.3.5[39]. The gene ontology (GO) enrichment was assessed by a Bioconductor package, GOstats (ver. 2.40.0)[40]. The R packages (ver. 3.4.1) (http://www.r-project.org) were used to plot a heatmap, compute the Pearson's correlation coefficients and P-values, and group the data by K-means clustering. Significantly enriched ChIP-seq regions, called peaks, were identified by MACS2 program (ver. 2.1.0)[38] with minimum false discovery rate (*q* value) cutoff of 0.05. The peak overlapping between samples was examined by DiffBind-2.2.3[41]. Gviz-1.18.2[42] visualized the location of the peaks on chromosomes.

**RNA preparation and sequencing.** Total RNA was isolated by hot-phenol method. The mRNA-seq libraries were prepared using NEXTflex Rapid Directional mRNA-Seq Kit (Bioo Scientific) and sequenced on the HiSeq2500 sequencer (Illumina). The sequences were aligned to the *S. cerevisiae* transcriptome using HISAT2–2.0.4[43]. The sequence reads were counted by StringTie-1.3.0[44] and fragments per kilobase of transcript per million mapped reads (FPKM) was used as a normalized expression level. The downstream analysis such as drawing heatmap and violin plot, and permutation test, was done by using R packages (ver. 3.4.1).

**Reporting summary.** Further information on research design is available in the Nature Research Reporting Summary linked to this article.

## Data availability

The ChIP-seq and RNA-seq data sets that support the findings of this study have been deposited in the Gene Expression Omnibus with the accession code GSE121763 (https://www.ncbi.nlm.nih.gov/geo/query/acc.cgi?acc=GSE121763). All other relevant data supporting the key findings of this study are available within the article and its Supplementary Information files or from the corresponding author upon reasonable request. The source data underlying Figs. 1, 3a–d, 3 g, 3 h, 4 g, and 5c, and Supplementary Figs. 1, 3b–d, 4d, 5c, and 6a are provided as a Source Data file. A reporting summary for this Article is available as a Supplementary Information file.

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

## Acknowledgements

We thank Stephen Buratowski (Harvard Medical School), Jerry Workman (Stowers Institute for Medical Research), and all members of the Kim lab for helpful advice and discussions. This work was supported by grants from the National Research Foundation (NRF-2017M3A9B5060887, NRF-2017M3A9G7073033, NRF-2017M3C9A5029980, and NRF-2019R1A5A6099645) to T.K. and NRF-2014M3C9A3064548 to T.-Y. R.

## Author contributions

S.D.H., T.-Y. R., and T.K. designed the project; S.D.H. and M.Y.K. performed most of experiments; S.D.H., I.J., and S.H. analyzed the data; J.H.K., B.B.L., I.J., M.K.L., and J.-T. H. contributed to the data analyses; S.D.H., S.H., T.-Y.R., and T.K. wrote the manuscript. All authors discussed the results and commented on the manuscript.

## Additional information

**Competing interests:** The authors declare no competing interests.

