## [Peer Review File · Nature Communications]

Reviewers' comments:

Reviewer #1 (Remarks to the Author):

Histone H4-specific deacetylation by Hda1C within active coding regions suppresses nucleosome instability

So Dam Ha, Seokjin Ham, Min Young Kim, Ji Hyun Kim, Insoon Jang, Bo Bae Lee, Jin-Taek Hwang, Tae-Young Roh, TaeSoo Kim

Ha et al. describe a role of the Hda1 complex in histone H4 deacetylation of active coding genes in budding yeast using genome-wide analyses i.e. ChIP-seq to study global effects on histone acetylation in the *hda1Δ* mutant and to study global Hda1 binding profile in wild-type. The main findings in this manuscript are that the loss of HDA1 causes hyper-acetylation of histone H4, not histone H3, within active coding regions partially through Arb2 domain of Hda1 in transcription dependent manner.

Overall, the data from this manuscript generally support their conclusions; however there are some points which need to be addressed.

Points of concern

1. The ChIP-seq analysis of histone acetylation should be normalized by histone H3 to rule out the possibility of acetylation increase due to the increase of nucleosome occupancy. Although the G5 in Fig 2b., which shows the strongest increase of H4 acetylation in the loss of HDA1, shows the decrease of H3 occupancy in the *hda1Δ* mutant (Fig 6a.), the global pattern of histone acetylation changes normalized by H3 changes should be analyzed.
2. The authors suggest that the Hda1C preferentially deacetylates histone H4 via a Tup1-independent mechanism, however, relative H4ac (*tup1Δ*/WT) in Fig 2a. seems that Tup1 also has a role in H4 deacetylation in part of highly transcribed genes (also found in Fig S2c.). Thus, more detailed analysis of acetylation changes in the *hda1Δ* and *tup1Δ* is required, such as to divide into Hda1-specific, Tup1-specific and common genes. Data which directly shows an independency of Tup1 at these genes, for example Tup1 ChIP, is also recommended.
3. The authors mention that "Furthermore, this binding was correlated with increases in H4 acetylation in *hda1Δ* cells (Fig. 4c).", however, the authors did not present correlation coefficient. It is required to present figures directly show the correlation between the Hda1 binding and the H4ac changes in the *hda1Δ* mutant.
4. In Fig 6a. the authors showed relative H3 patterns in the *hda1Δ* and the *set2Δ* mutants. To make a more detailed comparison and to clarify whether the Hda1C and Set2 differentially function

in regulating nucleosome occupancy, relative H3 pattern of Hda1-specific, Set2-specific, and common genes (Fig 5g.) should be analyzed to investigate the pattern of nucleosome occupancy changes in the *hda1Δ* and the *set2Δ* mutants where Hda1 and Set2 deacetylates histones H4.

5. In order to suggest the negative role of Hda1 in RNAPII elongation through its activity of histone H4 de-acetylation, RNAPII enrichment where H4ac is increased in the *hda1Δ* should be investigated using RNAPII ChIP in the *hda1Δ* mutant..

6. Finally, how about histone exchange in *hda1Δ* mutant? Is there any genetic interaction with histone chaperones such as Asf1 or Hir complex? The Set2-Rpd3 usually prevents the assembly of newly synthesized histones at infrequently transcribed genes. How about Hda1C? Is there any contribution by Hda1c to preserve epigenetic information like Set2-Rpd3 during highly transcribed genes?

Minor point

1. Related Fig 2b: Because the authors points that the function of the Hda1C in regulating histone H4 acetylation is in transcription dependent manner, the transcription level of each five clustered groups in Fig 2b. should also be analyzed.

2. Related Fig 3b: The authors mentioned that Hda1 binding is independent of Set1p and Set2p, however, it seems that the Hda1p binding in the *set1Δ set2Δ* mutant is slightly decreased.

3. Related Fig S6c and S6d: Please indicate whether “H3ac” and “H4ac” is relative value or enrichment in wild-type.

4. The authors suggest that the Hda1C also has a role in histone H3 acetylation at less active and inducible genes to delay the kinetics of gene expression upon induction. It would be better if the authors explain the in vivo meaning or benefit of delaying the kinetics upon environmental change in discussion part.

Reviewer #2 (Remarks to the Author):

This manuscript describes a new function for the Hda1C histone deacetylase complex. The authors demonstrate that Hda1 is recruited to the ORFs of highly transcribed genes where it deacetylates histone H4 and maintains nucleosomal occupancy. This is a significant finding that will be of widespread interest, however at present there are some major issues with the manuscript.

(i) To my mind, the most significant problem is a lack of mechanistic insight. It is not clear how Hda1 is recruited to transcribed regions. The authors suggest that Hda1 localisation to gene sequences is partially dependent upon RNA, however these data are not at all convincing and do not demonstrate any association with nascent transcripts. They also show by co-IP that Hda1 interacts with a subunit of RNA pol II however they do not show that interaction is specific to elongating pol II. As such the claim that Hda1 likely binds via an interaction with elongating Pol II and/or nascent transcripts is not justified. Furthermore, the authors' model does not explain how Hda1 discriminates between gene classes.

(ii) As indicated above, some of the data are not convincing. Importantly, the CHIP data described in Fig 1, Fig 3 and Fig 4f are derived from only two biological repeats. It is not possible to calculate standard deviation error bars based on two repeats. The experiments must be performed with a minimum of three repeats. Proper tests of significance are lacking in many places in the manuscript and, as a result, many of the conclusions are not warranted at this stage. For example, it is not clear that loss of HDA2 or HDA3 results in any significant reduction to Hda1 recruitment to YEF3 or PMA1. Similarly, are the decreases in Hda1 recruitment observed upon RNase treatment significant? If so why does RNase treatment reduce Hda1 occupancy at promoters?

(iii) How many biological repeats were used to generate the CHIP-seq data presented in Fig2? This information is not given in the figure legend or the materials and methods.

(iv) The levels of Hda1 and Hda1-delta Arb2 mutant should be quantified and the number of biological repeats used for this analysis should be specified. Importantly, the level of the mutant (Arb2 delta) Hda1 looks to be somewhat reduced and so it is possible that this is the explanation for the apparent reduction in the level of recruitment of this mutant.

(v) Figure 4 does not demonstrate that elongating RNA pol II recruits Hda1. The only interaction that is shown is with Rpb3 which is not specific to the elongating form of PolIII. The details of the numbers of repeats performed for Fig 4e should be indicated.

(vi) Since it is proposed that deletion of the Arb2 domain reduces recruitment of Hda1 then loss of this domain should also reduce the interaction with PolIII if this is a significant interaction.

(vii) The authors conclude that loss of Hda1 does not increase cryptic transcription based on analysis of only two genes (PCA1 and STE11) and H4 acetylation levels of STE11 are not Hda1 dependent. They should examine genes where nucleosomal occupancy is Hda1-dependent (ie the G4 and G5 genes described in Fig 6a) for increased levels of cryptic and antisense transcripts. Given that Hda1c

suppresses H4 acetylation levels and maintains nucleosomal occupancy in transcribed regions the lack of increased cryptic transcription in an *hda1* delete is puzzling.

The inclusion of data described in Fig6 c-f which relates to the function of Hda1 at promoters is odd and adds little to the main findings of this manuscript.

Reviewer #3 (Remarks to the Author):

Ha et al. investigate the effect of the Hda1 histone deacetylase complex on histone H3 and H4 acetylation levels and patterns in yeast. Hda1C has well established connections to Tup1 and transcriptional repression at promoters through regulating H3 acetylation levels. The authors provide evidence for a new role of Hda1C on coding regions. Through ChIP and ChIP-seq experiments, the authors find that deletion of HDA1 increases H4 acetylation on coding regions of active genes. This observation contrasts with the H3-specific effects observed at the promoters and coding regions of repressed genes. The authors also report that the increase in H4ac is positively correlated with Pol II density and argue that Hda1C binds to Pol II on coding regions. Consistent with this observation, the levels of Hda1 occupancy correlate with Pol II density. The authors present ChIP data to support the conclusion that Hda1C and Set2 function in different pathways to control genic H4ac levels. They show that H3 levels are reduced in the absence of Hda1 and conclude that Hda1C is important for nucleosome stability. Finally they provide RNA-seq data supporting a role for Hda1C in repressing genes that respond to changes in carbon source and link this effect to a suppression of H3 acetylation levels.

The primary contribution of this work is the finding of a new pathway for controlling histone acetylation levels on active genes. This pathway seems to be distinct, yet overlapping at some genes, with the well-studied Rpd3S-Set2 pathway. The finding that Hda1C is important for fine tuning transcriptional responses in the context of dynamic environmental conditions is also interesting. Other conclusions from this work are not sufficiently supported by the data and the authors have overlooked a previous study that demonstrated occupancy of Hda1 on coding regions. Finally, the paper is weakened by insufficient repetitions, a lack of statistical analysis, missing controls, and a tendency for over-interpretation.

Specific points:

1. Throughout the paper, little information is given about statistical significance. The qRT-PCR experiments are missing p-values and report data from only two biological replicates (three is standard). The methods are thin on details, especially in the section on ChIP-seq. It is unclear how many biological replicates were performed for the genomic studies, how the data were normalized, and whether spike-in controls were applied. The use of spike-in controls to measure changes in protein occupancy is now standard and is especially important when reporting global changes in histone levels.

2. The discovery of Hda1 occupancy on coding regions by ChIP is not new and the authors should have cited work from the Hinnebusch lab, which also showed transcription-dependent recruitment of Hda1-myc on the coding region of a gene (Mol Cell 2010 vol. 39, pp. 234-246). Putting this concern aside, the authors' demonstration of Hda1 occupancy on active genes genome-wide is a nice contribution.

3. Figure 2--Were the H3ac and H4ac levels normalized to total H3? What is meant by "relative H3ac" or "relative H4ac"? Normalization is critical because the authors also argue for effects of deleting Hda1 on nucleosome occupancy. The authors conclude that there is an increase in H4ac at promoters in the *tup1* mutant. This is not evident from the heat map. A different representation of the data is required.

4. Figure 2--The authors conclude that changes in acetylation levels correlate with Pol II density. This is not clear from the heat maps. A statistical measure of correlation should be provided and this analysis should be extended to the G1-5 groups. What is the source of the Rpb3 ChIP-seq data? If these are published data, a citation is needed. If these are new data, more details are needed in the Methods.

5. Figure 3—A control is needed to demonstrate that the Hda1-myc protein is active. Details are needed about this tagged protein. Is the tag on the N- or C-terminus? The authors argue that occupancy of Hda1-myc is higher on active genes than less active genes. Given the focus on coding regions, it is surprising that they did not show occupancy of Hda1-myc on the coding region of the inactive *TKL2* gene. Also, the data in panels a and b appear to be redundant and can be combined.

6. Figure 3—The authors address the mechanism of Hda1 recruitment and provide data to suggest that the two other complex members, Hda2 and Hda3, have a slight effect on Hda1 occupancy (p-values would be helpful here). They then show that Hda1-myc occupancy is reduced if the chromatin is RNase-treated. From this, they conclude that Hda1 recruitment involves RNA binding possibly by Hda1C subunits. This seems like a leap in logic. Just because deletions in Hda2/Hda3 and RNase-treatment lead to a reduction in Hda1 occupancy does not mean these two mechanisms are related.

A connection could be tested by asking if RNase treatment exacerbates or has no additional effect on Hda1 occupancy in a *hda2 hda3* double mutant.

7. Figure 3—The authors provide convincing evidence that the Arb2 domain of Hda1 is important for Hda1-myc recruitment. However the labeling in panel F (far right panels) is confusing, unless they are expressing the Arb2 domain alone. Also, what is the red box in panel D?

8. Figure 4b and c (lines 192-193)--By comparing the heat maps of the Hda1-myc and H4ac ChIP-seq experiments, the authors conclude a correlation. This is not evident from the heat maps. A statistical comparison is needed.

9. Figure 4d—A trace of a repressed gene would be a nice control here.

10. Figure 4e---The discovery of a protein-protein interaction between Hda1 and Rpb3 is potentially interesting. However, several yeast proteomic studies have been conducted at this point and, according to the summary of these studies in SGD, this interaction has not been reported previously. In contrast, the Paf1 control interaction has been observed many times. The paper lacks a sufficient description of the methods for this experiment. In addition, to bolster this unexpected result, the authors should consider providing additional experimental support. For example, is the interaction specific to a particular phosphorylation state of Pol II? Can it be detected with purified proteins? Is the Arb2 domain required? Have the authors considered the possibility that Hda1C association with active genes correlates with Pol II density because nucleosomes are remodeled in the wake of Pol II and serve as more effective binding targets for Hda1C? Similar models have recently been proposed by the Howe group for FACT.

11. Figure 4h---Why are these scales different from those in Figure 1a?

12. Figure 5---Panel c requires statistical analysis. The source of the H3K36me data should be cited or, if done by the authors, the methods need to be better described. The GO analysis adds little informational content.

13. Figure S5 requires positive controls.

14. Figure 6—The interpretation of the results is overstated. On line 259 and elsewhere, the authors conclude that “Hda1C suppresses nucleosome instability and partially inhibits elongation”. The authors measure H3 occupancy by CHIP-seq but do not measure nucleosome occupancy (by MNase-seq, for example). They also do not measure nucleosome stability. The conclusion that Hda1C inhibits elongation is based solely on genetic suppression of a *bur1* mutant. This is not sufficient evidence for such a strong mechanistic interpretation.

15. Figure 6e and 6f---These figures need a better description. On first read, the results appear at odds with those shown in Figure 1.

16. Supplementary Table 1—Auxotrophies should be unambiguous. It is not difficult to distinguish *lys2Δ* from *LYS2* using growth assays.

17. Throughout the paper--- The authors word choice of “active” vs “inactive” overstates differences between the genes examined. Many genes in the “inactive” group are transcribed at low levels. The authors seem to use the term “active” for only the most highly expressed genes.

We thank the reviewers for their insightful comments on our manuscript and we have addressed them accordingly. We hope that the editor and all reviewers will now find the revised manuscript ready for publication in Nature Communications.

Reviewer #1 (Remarks to the Author):

Ha et al. describe a role of the Hda1 complex in histone H4 deacetylation of active coding genes in budding yeast using genome-wide analyses i.e. ChIP-seq to study global effects on histone acetylation in the *hda1Δ* mutant and to study global Hda1 binding profile in wild-type. The main findings in this manuscript are that the loss of HDA1 causes hyper-acetylation of histone H4, not histone H3, within active coding regions partially through Arb2 domain of Hda1 in transcription dependent manner. Overall, the data from this manuscript generally support their conclusions; however there are some points which need to be addressed.

We thank the reviewer for recognizing the significance of our findings and for the helpful comments.

Points of concern

1. The ChIP-seq analysis of histone acetylation should be normalized by histone H3 to rule out the possibility of acetylation increase due to the increase of nucleosome occupancy. Although the G5 in Fig 2b., which shows the strongest increase of H4 acetylation in the loss of HDA1, shows the decrease of H3 occupancy in the *hda1Δ* mutant (Fig 6a.), the global pattern of histone acetylation changes normalized by H3 changes should be analyzed.

We completely agree with the reviewer and have analyzed histone acetylation patterns normalized with total H3 contents. The patterns of histone acetylation were compatible to those of the original data. In the revised manuscript, all ChIP-seqs for histone acetylation were normalized with total histone H3 levels.

2. The authors suggest that the Hda1C preferentially deacetylates histone H4 via a Tup1-independent mechanism, however, relative H4ac (*tup1Δ*/WT) in Fig 2a. seems that Tup1 also has a role in H4 deacetylation in part of highly transcribed genes (also found in Fig S2c). Thus, more detailed analysis of acetylation changes in the *hda1Δ* and *tup1Δ* is required, such as to divide into Hda1-specific, Tup1-specific and common genes. Data which directly shows an independency of Tup1 at these genes, for example Tup1 ChIP, is also recommended.

As the reviewer suggested, we have analyzed H4 acetylation patterns of *tup1Δ*. A metagene analysis of all genes indicated that loss of Hda1 strongly increased H4 acetylation within coding regions. In contrast, *tup1Δ* showed a small increase in H4 acetylation at promoters. A further analysis based on RNA Pol II levels indicated that top 20% and bottom 20% of yeast genes tend to have a slightly increased H4 acetylation in *tup1Δ* (Fig. R1). At this point, we don't exactly understand how Tup1 regulates H4 acetylation at these genes or whether this is a direct effect or not, and so decided not to include this data in the manuscript and change the text.

Fig. R1. H4 acetylation patterns in *tup1Δ*.

3. The authors mention that “Furthermore, this binding was correlated with increases in H4 acetylation in *hda1Δ* cells (Fig. 4c).”, however, the authors did not present correlation coefficient. It is required to present figures directly show the correlation between the Hda1 binding and the H4ac changes in the *hda1Δ* mutant.

As requested, we have analyzed the correlation between Hda1 binding and H4 acetylation changes. For the highly transcribed genes (top 25%), Hda1 binding and H4ac increase upon HDA1 deletion showed a statistically significant correlation (Supplementary Fig. 4b).

4. In Fig 6a. the authors showed relative H3 patterns in the *hda1Δ* and the *set2Δ* mutants. To make a more detailed comparison and to clarify whether the Hda1C and Set2 differentially function in regulating nucleosome occupancy, relative H3 pattern of Hda1-specific, Set2-specific, and common genes (Fig 5g.) should be analyzed to investigate the pattern of nucleosome occupancy changes in the *hda1Δ* and the *set2Δ* mutants where Hda1 and Set2 deacetylates histones H4.

*During the revision of this manuscript, we performed additional ChIP-seqs for histone H3 and H4 acetylation including S. pombe spike-in control. We carefully analyzed H3 occupancy in both WT and *hda1Δ* and found that histone H3 levels were slightly increased at actively transcribed genes where H4 acetylation is strongly increased in *hda1Δ*. This pattern was also seen in the original dataset (Fig. R2).*

We realized that inadvertently there was a mislabeling of a data resulting in misinterpretation of the results. We apologize for this mistake and thank the reviewer for pointing it out. We double-checked all the data used in the revised version of this manuscript.

Fig. R2. Histone occupancy in *hda1Δ* or *set2Δ*.

*As the reviewer suggested, we have analyzed the nucleosome occupancy of Hda1-specific, Set2-specific, or common genes from Fig. 5f. Loss of Set2 had reduced nucleosome occupancy at Set2-specific genes and common genes but *hda1Δ* showed the opposite patterns (Figure R2). At this point, we don't exactly understand why histone density is slightly increased when histone H4 is hyperacetylated in *hda1Δ*. We therefore decided to move the data on nucleosome occupancy to Supplementary Fig. 5 and have changed the title and the text accordingly.*

Except for this, all data were highly reproducible, and we continue to believe that Hda1C specifically deacetylates histone H4 at highly active genes.

5. In order to suggest the negative role of Hda1 in RNAPII elongation through its activity of histone H4 de-acetylation, RNAPII enrichment where H4ac is increased in the *hda1Δ* should be investigated using RNAPII ChIP in the *hda1Δ* mutant.

As the reviewer suggested, we have performed Rpb3 ChIP in WT and *hda1Δ*. Although H4 acetylation was strongly increased within coding regions of YEF3 and PMA1, Rpb3 crosslinking was similar in both WT and *hda1Δ* (Fig. R3).

Fig. R3. RNA Pol II occupancy in WT and *hda1Δ*.

Previous studies also showed no effect of hyperacetylation within coding regions on RNA Pol II occupancy. For example, loss of Set3 HDAC caused increased acetylation in 5' transcribed regions of YEF3 and PMA1 but had no effect on transcript levels and Rpb3 occupancy (Kim and Buratowski, *Cell*, 2009). Furthermore, many chromatin regulators had a marginal effect on global transcript levels in a steady-state condition (Lenstra et al., *Molecular Cell*, 2011). Instead, they likely play an important role in regulating the kinetics of gene induction or repression upon environmental changes (Alejandro-Osorio et al., *Genome Biology*, 2009; Kim et al., *Cell*, 2012; Kim et al., *Nature Communications*, 2016; Lee et al., *Nucleic Acids Research*, 2018).

We chose not to add this result to the paper but can do so if the editor and the reviewer think it important.

6. Finally, how about histone exchange in *hda1Δ* mutant? Is there any genetic interaction with histone chaperones such as Asf1 or Hir complex? The Set2-Rpd3 usually prevents the assembly of newly synthesized histones at infrequently transcribed genes. How about Hda1C? Is there any contribution by Hda1c to preserve epigenetic information like Set2-Rpd3 during highly transcribed genes?

We thank the reviewer for this interesting suggestion. Since Hda1C is important for H4 deacetylation at highly active coding regions, it could affect histone exchange during transcription elongation by RNA Pol II. However, although it is an important question, we think that elucidating the function of Hda1C in histone exchange would be beyond the scope of this study because in this manuscript, we have focused on a novel function of Hda1C in H4-specific deacetylation within highly transcribed genes.

To briefly test the function of Hda1C on histone exchange, we performed a ChIP to determine H3K56 acetylation, a mark connected to histone exchange. Consistent with the main finding of this study, loss of Hda1C had no strong effect on H3K56 acetylation within coding regions of YEF3 and PMA1 (Fig. R4) suggesting that Hda1C may not be involved in histone exchange mediated by H3K56 acetylation.

Fig. R4. H3K56 acetylation levels in WT and *hda1Δ*.

Minor point

1. Related Fig 2b: Because the authors points that the function of the Hda1C in regulating histone H4 acetylation is in transcription dependent manner, the transcription level of each five clustered groups in Fig 2b. should also be analyzed.

We have analyzed RNA Pol II occupancy for the 5 groups in Fig. 2d and found that group 4 and 5 having highest levels of H4 acetylation upon HDA1 deletion tend to exhibit higher levels of RNA Pol II. This data now appears in Supplementary Fig. 4c.

2. Related Fig 3b: The authors mentioned that Hda1 binding is independent of Set1p and Set2p, however, it seems that the Hda1p binding in the set1Δ set2Δ mutant is slightly decreased.

We have performed Hda1-ChIPs in double mutant with three biological replicates. Although this is statistically significant, Hda1 binding was reduced only by about 10% in set1Δset2Δ (Fig. R5). We decided to keep the original data and changed the text accordingly.

Fig. R5. Hda1 occupancy in WT and set1Δset2Δ.

3. Related Fig S6c and S6d: Please indicate whether “H3ac” and “H4ac” is relative value or enrichment in wild-type.

This has been corrected.

4. The authors suggest that the Hda1C also has a role in histone H3 acetylation at less active and inducible genes to delay the kinetics of gene expression upon induction. It would be better if the authors explain the in vivo meaning or benefit of delaying the kinetics upon environmental change in discussion part.

We have added the sentences describing the importance of regulation of gene expression dynamics upon environmental changes.

Reviewer #2 (Remarks to the Author):

This manuscript describes a new function for the Hda1C histone deacetylase complex. The authors demonstrate that Hda1 is recruited to the ORFs of highly transcribed genes where it deacetylates histone H4 and maintains nucleosomal occupancy. This is a significant finding that will be of widespread interest, however at present there are some major issues with the manuscript.

We thank the reviewer for the positive remarks on our manuscript.

(i) To my mind, the most significant problem is a lack of mechanistic insight. It is not clear how Hda1 is recruited to transcribed regions. The authors suggest that Hda1 localisation to gene sequences is partially dependent upon RNA, however these data are not at all convincing and do not demonstrate any association with nascent transcripts. They also show by co-IP that Hda1 interacts with a subunit of

RNA pol II however they do not show that interaction is specific to elongating pol II. As such the claim that Hda1 likely binds via an interaction with elongating Pol II and/or nascent transcripts is not justified.

We respectfully disagree with the reviewer on this comment. Hda1C was previously known to be recruited to inactive promoters by Tup1 corepressor and preferentially deacetylate histone H3 and H2B to repress transcription (Wu et al., Molecular Cell, 2001). However, we showed here that this complex was also targeted to actively transcribed genes to specifically deacetylate histone H4. Our data clearly showed that Hda1C binds to target genes in a transcription-dependent manner (Fig. 4). Furthermore, we also showed for the first time that the Arb2 domain of Hda1 is crucial for its association to chromatin (Fig. 3g and Fig. 4b). During the revision of this paper, we tested whether the Arb2 domain was required for the interaction between RNA Pol II and Hda1. Consistent with our ChIPs and ChIP-seq data, loss of the Arb2 domain resulted in a strong reduction of the interaction (Fig. 4f). The interaction between the Arb2 domain and RNA Pol II provides a novel mechanistic insight of how Hda1C is targeted to active genes.

*For the interaction with elongating RNA Pol II, we have done Hda1-myc ChIP in WT and *ctk1Δ* and found that there was no strong difference in Hda1 binding (Supplementary Fig. 3d). This result suggests that Hda1C may not directly recognize phosphorylated serine 2 of C-terminal domain of Rpb1.*

*For the interaction with nascent RNA transcripts, we showed that Hda1 binding was decreased by RNase treatment (Fig. 3c). As the reviewer 3 suggested, we have carried out Hda1 ChIP in *hda2Δhda3Δ* mutant with/without RNase treatment. Again, RNase treatment reduced Hda1 binding in WT, this was not seen in *hda2Δhda3Δ* mutant (Fig. 3d). Although additional experiments, for example CLIP-seqs, are required, this result suggests that Hda1 recruitment is partially dependent on the interaction with nascent RNA transcripts.*

Furthermore, the authors' model does not explain how Hda1 discriminates between gene classes.

As the reviewer pointed out, at this point, we don't exactly understand how Hda1C differentially deacetylates histone H3 or H4 at distinct loci. This is an important question, and as we mentioned in discussion section, an interesting possibility is via post-translational modifications of Hda1C since Hda1 and Hda3 can be either phosphorylated or sumoylated (Lewicki et al., J. Proteomics, 2015; Yeast Genome Database). Whether these modifications affect the specificity of Hda1C will be interesting.

(ii) As indicated above, some of the data are not convincing. Importantly, the ChIP data described in Fig 1, Fig 3 and Fig 4f are derived from only two biological repeats. It is not possible to calculate standard deviation error bars based on two repeats. The experiments must be performed with a minimum of three repeats. Proper tests of significance are lacking in many places in the manuscript and, as a result, many of the conclusions are not warranted at this stage. For example, it is not clear that loss of HDA2 or HDA3 results in any significant reduction to Hda1 recruitment to YEF3 or PMA1. Similarly, are the decreases in Hda1 recruitment observed upon RNase treatment significant? If so why does RNase treatment reduce Hda1 occupancy at promoters?

*We have repeated ChIPs for the key results with three biological repeats and added new data including the statistical analysis in the revised manuscript. Importantly, we have performed additional ChIP-seqs using normalizing *S. pombe* spike-in control. For Hda1-myc, Hda1-myc (Δ Arb2), H4 acetylation, and histone H3, we have done four ChIP-seqs and the data are highly reproducible (ex. Fig. 2a and 2b for H4 acetylation)*

*ChIPs with three biological replicates indicated that Hda1 occupancy was significantly reduced in *hda2Δhda3Δ*. In addition, this reduction was not observed in double mutant when RNase was treated (Fig. 3b-3d).*

(iii) How many biological repeats were used to generate the ChIP-seq data presented in Fig2? This information is not given in the figure legend or the materials and methods.

We apologize for not providing the details of ChIP-seqs and have added the information in the figure legend. Original ChIP-seqs were from two biological replicates. As written in our response above, during the revision, we have performed additional ChIP-seqs having spike-in control for Hda1 binding, histone H3, and H4 acetylation with two biological replicates.

(iv) The levels of Hda1 and Hda1-delta Arb2 mutant should be quantified and the number of biological repeats used for this analysis should be specified. Importantly, the level of the mutant (Arb2 delta) Hda1 looks to be somewhat reduced and so it is possible that this is the explanation for the apparent reduction in the level of recruitment of this mutant.

We have performed additional western blotting with three biological repeats and found that the level of Hda1 protein was reduced by only about 13% (Fig. R6). This information has been added into the text.

Fig. R6. Protein levels of WT Hda1 and ΔArb2 mutant.

(v) Figure 4 does not demonstrate that elongating RNA pol II recruits Hda1. The only interaction that is shown is with Rpb3 which is not specific to the elongating form of PolII. The details of the numbers of repeats performed for Fig 4e should be indicated.

We have performed two independent experiments to determine the interaction between Rpb3 and Hda1. This information has been added in the figure legend. During the revision, we have performed co-immunoprecipitation assays with two biological replicates to examine the interaction between Rpb3 and WT Hda1 or the Arb2 domain mutant.

As the reviewer pointed out, this result does not indicate that Hda1 interacts with elongating RNA Pol II. However, our results clearly demonstrated that this complex is targeted to highly active genes by active RNA Pol II (Figure 4a, 4b, and 4g), so we have replaced “elongating RNA Pol II” to “active RNA Pol II” in the manuscript.

(vi) Since it is proposed that deletion of the Arb2 domain reduces recruitment of Hda1 then loss of this domain should also reduce the interaction with PolII if this is a significant interaction.

We thank the reviewer for pointing out this interesting issue and have performed co-immunoprecipitation assay to monitor the interaction between Rpb3 and WT Hda1 or the Arb2 domain mutant. Again, WT Hda1 strongly interacted with Rpb3 but this interaction was significantly reduced when the Arb2 domain was deleted. This result has been added in Figure 4f.

(vii) The authors conclude that loss of Hda1 does not increase cryptic transcription based on analysis of only two genes (PCA1 and STE11) and H4 acetylation levels of STE11 are not Hda1 dependent. They should examine genes where nucleosomal occupancy is Hda1-dependent (ie the G4 and G5 genes described in Fig 6a) for increased levels of cryptic and antisense transcripts. Given that Hda1c suppresses H4 acetylation levels and maintains nucleosomal occupancy in transcribed regions the lack of increased cryptic transcription in an hda1 delete is puzzling.

As the reviewer suggested, we have analyzed the ratio between 3' transcript levels (80-100%) and 5' ones (1-20%) using RNA-seq data to see if cryptic transcripts are produced in hda1Δ. All five groups in Fig. 2d showed no significant difference in the ratio suggesting that Hda1C may have no effect on cryptic initiation (Fig. R7).

Fig. R7. Effect of hda1Δ on cryptic transcription.

As written in our reply to reviewer 1 and 3, careful reanalysis of histone occupancy indicates that loss of Hda1 causes a slight increase of histone occupancy within coding regions where H4 is hyperacetylated. This may explain why cryptic transcripts are not seen in hda1Δ.

Alternatively, Hda1C may regulate cryptic promoters activated upon environmental changes. Previous studies have shown that the Set2-Rpd3S HDAC pathway represses cryptic transcription by deacetylating histones within coding regions (Carrozza et al., Cell, 2005; Li et al., Genes and Development, 2007). However, ~50% of genes having increased acetylation within coding regions upon loss of the pathway had no detectable cryptic or antisense transcripts (Li et al., Genes and Development, 2007). Instead, these genes tend to have environmentally induced non-coding RNA transcripts (Kim et al., Nature Communications, 2016).

The inclusion of data described in Fig6 c-f which relates to the function of Hda1 at promoters is odd and adds little to the main findings of this manuscript.

Although Hda1C is known to regulate H3 and H2B acetylation at inactive promoters, its function on transcription has not been fully elucidated. Our data indicate that this function is important to delay the kinetics of gene induction upon environmental changes. As the reviewer 1 and 3 pointing it out as interesting findings, we decided to keep this data in the revised manuscript and have added a bit more explanation.

Reviewer #3 (Remarks to the Author):

Ha et al. investigate the effect of the Hda1 histone deacetylase complex on histone H3 and H4 acetylation levels and patterns in yeast. Hda1C has well established connections to Tup1 and transcriptional repression at promoters through regulating H3 acetylation levels. The authors provide evidence for a new role of Hda1C on coding regions. Through ChIP and ChIP-seq experiments, the authors find that deletion of HDA1 increases H4 acetylation on coding regions of active genes. This observation contrasts with the H3-specific effects observed at the promoters and coding regions of repressed genes. The authors also report that the increase in H4ac is positively correlated with Pol II density and argue that Hda1C binds to Pol II on coding regions. Consistent with this observation, the levels of Hda1 occupancy correlate with Pol II density. The authors present ChIP data to support the conclusion that Hda1C and Set2 function in different pathways to control genic H4ac levels. They show that H3 levels are reduced in the absence of Hda1 and conclude that Hda1C is important for nucleosome stability. Finally they provide RNA-seq data supporting a role for Hda1C in repressing genes that respond to changes in carbon source and link this effect to a suppression of H3 acetylation levels.

The primary contribution of this work is the finding of a new pathway for controlling histone acetylation levels on active genes. This pathway seems to be distinct, yet overlapping at some genes, with the well-studied Rpd3S-Set2 pathway. The finding that Hda1C is important for fine tuning transcriptional responses in the context of dynamic environmental conditions is also interesting. Other conclusions from this work are not sufficiently supported by the data and the authors have overlooked a previous study that demonstrated occupancy of Hda1 on coding regions. Finally, the paper is weakened by insufficient repetitions, a lack of statistical analysis, missing controls, and a tendency for over-interpretation.

We appreciate the reviewers' positive evaluation and very detailed reading of the manuscript.

Specific points:

1. Throughout the paper, little information is given about statistical significance. The qRT-PCR experiments are missing p-values and report data from only two biological replicates (three is standard). The methods are thin on details, especially in the section on ChIP-seq. It is unclear how many biological replicates were performed for the genomic studies, how the data were normalized, and whether spike-in controls were applied. The use of spike-in controls to measure changes in protein occupancy is now standard and is especially important when reporting global changes in histone levels.

As written in our reply to Reviewer 1 and 2, during the revision of this manuscript, we have repeated

*the key results with three biological replicates and analyzed the statistical significance. In the revised manuscript, all ChIP-seqs for histone acetylation were normalized with total histone H3 levels. Furthermore, we also performed additional ChIP-seqs for Hda1 binding, H3, and H4 acetylation with two biological replicates. These new ChIP-seqs had spiked-in *S. pombe* chromatin as a normalizing control. We have added a bit more explanation in figure legend and methods.*

2. The discovery of Hda1 occupancy on coding regions by ChIP is not new and the authors should have cited work from the Hinnebusch lab, which also showed transcription-dependent recruitment of Hda1-myc on the coding region of a gene (Mol Cell 2010 vol. 39, pp. 234-246). Putting this concern aside, the authors' demonstration of Hda1 occupancy on active genes genome-wide is a nice contribution. *We apologize for not citing the paper from Hinnebusch lab and have added this paper in reference. This study showed Hda1 binding to ARG1 gene in a transcription-dependent manner. Our ChIP data on GAL genes also indicated that Hda1 bound to these genes only when they are actively transcribed by RNA Pol II (Figure 4g).*

3. Figure 2--Were the H3ac and H4ac levels normalized to total H3? What is meant by "relative H3ac" or "relative H4ac"? Normalization is critical because the authors also argue for effects of deleting Hda1 on nucleosome occupancy. The authors conclude that there is an increase in H4ac at promoters in the *tup1* mutant. This is not evident from the heat map. A different representation of the data is required. *We totally agree with the reviewer. In the revised manuscript, all ChIP-seqs for histone acetylation were normalized with total histone H3 levels.*

*As written in our response to Reviewer 1, we have analyzed H4 acetylation changes in WT and *tup1Δ* and found that there was a small increase of H4 acetylation at promoters. At this point, we don't exactly understand whether this is a direct effect or not. To focus on the main finding of this study, we decided to remove the sentence.*

4. Figure 2--The authors conclude that changes in acetylation levels correlate with Pol II density. This is not clear from the heat maps. A statistical measure of correlation should be provided and this analysis should be extended to the G1-5 groups. What is the source of the Rpb3 ChIP-seq data? If these are published data, a citation is needed. If these are new data, more details are needed in the Methods. *We have done this analysis and added new figures in Supplementary Fig. 4b and 4c. Also, the reference of the Rbp3 ChIP data has been added.*

5. Figure 3—A control is needed to demonstrate that the Hda1-myc protein is active. Details are needed about this tagged protein. Is the tag on the N- or C-terminus? The authors argue that occupancy of Hda1-myc is higher on active genes than less active genes. Given the focus on coding regions, it is surprising that they did not show occupancy of Hda1-myc on the coding region of the inactive *TKL2* gene. Also, the data in panels a and b appear to be redundant and can be combined. *The myc tag was fused to C-terminus of Hda1 and this information has been added in methods. To test whether Hda1-myc is active, we simply monitored H4 acetylation in untag control and Hda1-myc strains. Although *hda1Δ* showed a strong increase of H4 acetylation, this was not seen in Hda1-myc indicating that this protein is active (Fig. R8).*

Fig. R8. Hda1-myc is active.

*As requested, we have added ChIP data for *TKL2* ORF and reorganized Fig. 3 to avoid redundancy.*

6. Figure 3—The authors address the mechanism of Hda1 recruitment and provide data to suggest that the two other complex members, Hda2 and Hda3, have a slight effect on Hda1 occupancy (p-values

would be helpful here). They then show that Hda1-myc occupancy is reduced if the chromatin is RNase-treated. From this, they conclude that Hda1 recruitment involves RNA binding possibly by Hda1C subunits. This seems like a leap in logic. Just because deletions in Hda2/Hda3 and RNase-treatment lead to a reduction in Hda1 occupancy does not mean these two mechanisms are related. A connection could be tested by asking if RNase treatment exacerbates or has no additional effect on Hda1 occupancy in a *hda2 hda3* double mutant.

*To further confirm whether Hda2 and Hda3 affect Hda1 recruitment, we have repeated Hda1-myc ChIP in WT and *hda2Δhda3Δ* double mutant with three biological replicates. Loss of both Hda2 and Hda3 clearly reduced Hda1 binding and p-values have been added (Fig. 3b). Importantly, this reduction was not observed in *hda2Δhda3Δ* double mutant when RNase was treated further supporting that Hda1 recruitment is partially dependent on RNA binding of Hda2 and Hda3 (Fig. 3c and 3d).*

7. Figure 3—The authors provide convincing evidence that the Arb2 domain of Hda1 is important for Hda1-myc recruitment. However the labeling in panel F (far right panels) is confusing, unless they are expressing the Arb2 domain alone. Also, what is the red box in panel D?

This has been corrected. The red box indicates the low complexity region (from SMART; <http://smart.embl-heidelberg.de/>) and has been removed since it does not provide any information.

8. Figure 4b and c (lines 192-193)—By comparing the heat maps of the Hda1-myc and H4ac ChIP-seq experiments, the authors conclude a correlation. This is not evident from the heat maps. A statistical comparison is needed.

We have done this analysis and added new figures in Supplementary Fig. 4b and 4c.

9. Figure 4d—A trace of a repressed gene would be a nice control here.
*An inactive gene, *TKL2*, has been added.*

10. Figure 4e—The discovery of a protein-protein interaction between Hda1 and Rpb3 is potentially interesting. However, several yeast proteomic studies have been conducted at this point and, according to the summary of these studies in SGD, this interaction has not been reported previously. In contrast, the Paf1 control interaction has been observed many times. The paper lacks a sufficient description of the methods for this experiment. In addition, to bolster this unexpected result, the authors should consider providing additional experimental support. For example, is the interaction specific to a particular phosphorylation state of Pol II? Can it be detected with purified proteins? Is the Arb2 domain required? Have the authors considered the possibility that Hda1C association with active genes correlates with Pol II density because nucleosomes are remodeled in the wake of Pol II and serve as more effective binding targets for Hda1C? Similar models have recently been proposed by the Howe group for FACT.

In response to the first point, we have added the methods for co-immunoprecipitation assays in the method section and apologize for not including this in the original manuscript.

*As the reviewer pointed out, the interaction between Hda1C and RNA Pol II was completely unexpected. However, we had the same results from at least four independent experiments (Figure 4e and f). We think that the Arb2 domain directly interacts with RNA Pol II because deletion of the Arb2 domain resulted in a significant reduction of the interaction (Fig 4f). In contrast, serine 2 phosphorylation of RNA Pol II may not be required for this interaction as loss of *Ctk1* had almost no effect on Hda1 binding (Supplementary Fig. 3d).*

As the reviewer pointed out, we cannot exclude the possibility that nucleosome remodeling during transcription elongation could promote Hda1 binding to active coding regions. The Howe group proposed this model for FACT complex because there is no direct interaction between FACT and RNA Pol. However, we showed that Hda1C strongly interacts with RNA Pol II (Fig. 4e and 4f).

11. Figure 4h—Why are these scales different from those in Figure 1a?

This figure has been moved to Supplementary Fig.4d. In this figure, histone acetylation levels of WT were set to 1.

12. Figure 5---Panel c requires statistical analysis. The source of the H3K36me data should be cited or, if done by the authors, the methods need to be better described. The GO analysis adds little informational content.

*We have repeated this ChIP with three biological replicates and added two new genes showing a clear difference. The reference for H3K36me data has been added. In addition, we modified the GO analysis to emphasize the distinct function of Hda1 and Set2 and this has been moved to **Supplementary Fig. 5d**.*

13. Figure S5 requires positive controls.

We have been done this experiment with different chromatin regulators including Rpd3L, NuA4 HAT and NuA3 HAT. NuA3 complex contains two proteins, Nto1 and Pdp3, that recognize H3K36me3. As a positive control, Nto1 binding to H3K36me3 has been added.

14. Figure 6---The interpretation of the results is overstated. On line 259 and elsewhere, the authors conclude that “Hda1C suppresses nucleosome instability and partially inhibits elongation”. The authors measure H3 occupancy by ChIP-seq but do not measure nucleosome occupancy (by MNase-seq, for example). They also do not measure nucleosome stability. The conclusion that Hda1C inhibits elongation is based solely on genetic suppression of a bur1 mutant. This is not sufficient evidence for such a strong mechanistic interpretation.

*As written in our reply to Reviewer 1, we realized that there was a mistake in analysis of histone occupancy because of inadvertent mislabeling of a data. We apologize for this mistake. Careful reanalysis of histone occupancy indicates that loss of Hda1 results in a slight increase in histone levels. This was totally unexpected results but highly reproducible from four independent ChIP-seqs. These data now appear in **Supplementary Fig. 5** and we have changed the title of this manuscript and the text accordingly.*

*In addition, since bypassing the BUR1 requirement by loss of Hda1C is not sufficient to support that Hda1C negatively regulates elongation, we have changed the text as “H4 deacetylation by Hda1C may inhibit RNA Pol II elongation”. This figure has been moved to **Fig. 5h** and **Supplementary Fig. 5**.*

15. Figure 6e and 6f---These figures need a better description. On first read, the results appear at odds with those shown in Figure 1.

We have added a bit more explanation on this.

16. Supplementary Table 1---Auxotrophies should be unambiguous. It is not difficult to distinguish lys2Δ from LYS2 using growth assays.

This has been corrected.

17. Throughout the paper--- The authors word choice of “active” vs “inactive” overstates differences between the genes examined. Many genes in the “inactive” group are transcribed at low levels. The authors seem to use the term “active” for only the most highly expressed genes.

The term “active” has been replaced with “hyperactive” or “highly active/transcribed” throughout the paper.

Reviewers' comments:

Reviewer #1 (Remarks to the Author):

Ha et al. have added many new experiments to address the concerns raised in the initial round of review. Ha et al. also noticed that there is a mistake to interpret H3 occupancy calculation but they properly corrected the whole interpretations using them. Generally speaking, the revised one is stronger in its conclusions. However, Ha et al. did not reflect the results of the newly added data such as point 2 and 5 of rebuttal letter in the model figure. The observations that increased H3 occupancy by loss of HDA1, and no change of pol2 occupancy and H3K56Ac in PMA1 and YEF3 by HDA1 deletion strains suggest that there is a caveat to interpret the relationship between Hdac1C and transcription elongation or histone exchange. Ha et al. may tone-down these points and explain their model properly.

Minor point

In Discussion section, "The Set2-Rpd3S HDAC pathway slows elongation". Actually, those references didn't talk about slow down elongation.

In Methods section, "To generate Δ Arb2 strain in Fig. 3, " Δ Arb2 \rightarrow Δ arb2

In Methods section, "protease inhibitors (Pepstatin A 1 mM, Aprotinin 0.3 mM, Leupeptin 1 mM, PMSF 100 mM)": check each concentration of protease inhibitor. PMSF may be 1mM and other may have different dimensions.

In Sequencing analysis, you may add how to calculate spike in experiment more detail.

Reviewer #2 (Remarks to the Author):

In the revised version of this manuscript the authors have addressed many of the issues that were raised in the original review. As a result the manuscript is much improved, however there are some issues that remain.

The tendency to over interpretation is still present in places. For example, what is meant by the term 'active' RNA polymerase II? Presumably, active means transcribing? As outlined in the original review the authors do not demonstrate an interaction between elongating/transcribing RNA pol II and Hda1. The term 'active' seems to be an invention of the authors and should be removed or at least defined.

Another example is the interpretation of the RNase experiments. These experiments are consistent with an interaction with nascent transcripts but they do not in any way constitute proof. Alternative interpretations of these data are possible. Sentences such as that on page 4 "likely through an interaction with active RNA pol II and nascent RNA" should be re-written.

Fig 4g does not show that Hda1 travels with RNA pol II (as is claimed in the Figure legend), only that it can be ChIPed to GAL3 and GAL1 and that the level seems to be increased in galactose (although given the number of repeats whether this increase is statistically significant is not clear). Furthermore, there is still an Hda1 signal in the absence of any Rpb3 signal at GAL1 suggesting there may be an RNA pol II independent mode of recruitment.

The issue of biological repeats remains. Despite the original comments there are still places (eg Fig 3a and Fig 3h) in which standard deviation has been calculated from two biological repeats. This is not acceptable. Also the number of repeats is not explicitly specified for Fig1b or Fig4g. It should be.

Reviewer #3 (Remarks to the Author):

Ha et al. have done a nice job addressing previous reviewer comments and improving this manuscript. In addition to better describing experimental methods, repetitions, and statistical tests, the authors have performed key experiments in response to reviewer suggestions. Importantly, the authors have repeated ChIP-seq experiments with inclusion of a spike-in control. This normalization step has changed one of the previous conclusions. Instead of *hda1Δ* reducing H3 occupancy, the authors now report a slight elevation in H3 occupancy, which correlates with the lack of cryptic initiation in these cells. Furthermore, the authors have now normalized the H4ac ChIP-seq data to total H3, bolstering their arguments that H4ac is significantly elevated on active coding regions in *hda1Δ* cells. Other valuable additions, which provide some mechanistic insights, are the experiments in Figures 3D and 4F. These two experiments probe the mechanism of Hda1C recruitment to coding regions and thus raise the overall impact of this study. Taken together, the new experiments and controls, improved description of experimental methods and reproducibility, and changes to the text have significantly improved this manuscript. The result is a strong paper that will be of interest to many in the chromatin/transcription fields.

I have a few minor suggestions:

1. Page 8--- I recommend changing "binds directly" to "associates with" in the header. ChIP assays don't show direct DNA binding.
2. Figure 5C and top of page 13--- Can the authors provide a brief explanation for the choice of STT4 and HAP1 for these ChIP experiments? Elsewhere, PMA1 and YEF3 were the test genes.
3. Page 14, bottom--- The citations to Figures 2d and 2f are a bit confusing. Since those panels show normalized data, it's difficult to draw conclusions about total histone occupancy.
4. Figure 3 legend needs to be corrected. Panels h and i should be changed to g and h. Also, the sentence "The Arb2 domain is required for Hda1" is incomplete.
5. Why are *K. lactis* strains mentioned in the strain list? I thought the spike-in strain was *S. pombe*.

We thank the reviewers for their comments on our manuscript and we have addressed them accordingly. We hope that the editor and all reviewers will now find this revised manuscript ready for publication in Nature Communications.

Reviewer #1 (Remarks to the Author):

Ha et al. have added many new experiments to address the concerns raised in the initial round of review. Ha et al. also noticed that there is a mistake to interpret H3 occupancy calculation but they properly corrected the whole interpretations using them. Generally speaking, the revised one is stronger in its conclusions. However, Ha et al. did not reflect the results of the newly added data such as point 2 and 5 of rebuttal letter in the model figure. The observations that increased H3 occupancy by loss of HDA1, and no change of pol2 occupancy and H3K56Ac in PMA1 and YEF3 by HDA1 deletion strains suggest that there is a caveat to interpret the relationship between Hdac1C and transcription elongation or histone exchange. Ha et al. may tone-down these points and explain their model properly.

We thank the reviewer for the comments and have changed our model slide and the text accordingly.

Minor point

In Discussion section, "The Set2-Rpd3S HDAC pathway slows elongation". Actually, those references didn't talk about slow down elongation.

We think that the two references are correct. The paper from Carrozza et al. showed that this pathway contributed to repress cryptic promoters within ORFs. The paper from Keogh et al. showed that loss of this pathway bypassed the requirement of a positive elongation factor, Bur1. Mutants for this pathway also showed resistance to 6-AU and MPA that inhibit transcription elongation by reducing NTP pools. Furthermore, loss of Set2 restored RNA Pol II levels in bur2Δ mutant background indicating that the Set2-Rpd3S HDAC pathway negatively regulates elongation.

In Methods section, "To generate ΔArb2 strain in Fig. 3, " ΔArb2 -> Δarb2

The sentence has been changed as "To generate the mutant lacking the Arb2 domain in Fig. 3"

In Methods section, "protease inhibitors (Pepstatin A 1 mM, Aprotinin 0.3 mM, Leupeptin 1 mM, PMSF 100 mM)": check each concentration of protease inhibitor. PMSF may be 1mM and other may have different dimensions.

We thank the reviewer for pointing it out and this issue has been corrected.

In Sequencing analysis, you may add how to calculate spike in experiment more detail.

We have added a bit more explanation on this.

Reviewer #2 (Remarks to the Author):

In the revised version of this manuscript the authors have addressed many of the issues that were raised in the original review. As a result the manuscript is much improved, however there are some issues that remain.

We thank the reviewer for the comments.

The tendency to over interpretation is still present in places. For example, what is meant by the term 'active' RNA polymerase II? Presumably, active means transcribing? As outlined in the original review the authors do not demonstrate an interaction between elongating/transcribing RNA pol II and Hda1. The term 'active' seems to be an invention of the authors and should be removed or at least defined.

As requested, the term "active" in the sentence "the interaction between Hda1C and active RNA Pol II" has been removed.

Another example is the interpretation of the RNase experiments. These experiments are consistent with an interaction with nascent transcripts but they do not in any way constitute proof. Alternative interpretations of these data are possible. Sentences such as that on page 4 "likely through an interaction with active RNA pol II and nascent RNA" should be re-written.

The sentence has been modified.

Fig 4g does not show that Hda1 travels with RNA pol II (as is claimed in the Figure legend), only that it can be ChIPed to GAL3 and GAL1 and that the level seems to be increased in galactose (although given the number of repeats whether this increase is statistically significant is not clear). Furthermore, there is still an Hda1 signal in the absence of any Rpb3 signal at GAL1 suggesting there may be an RNA pol II independent mode of recruitment.

The sentence "Hda1 travels with RNA Pol II", has been changed. In addition, we have repeated ChIPs with three biological replicates and added new data including the statistical analysis.

As the reviewer pointed out, there might be an additional mechanism recruiting Hda1C since we saw an Hda1 signal at inactive genes, GAL genes and TKL2. This binding was completely gone when the Arb2 domain of Hda1 is deleted. We therefore think that the interaction between histones and the Arb2 domain contributes to Hda1 binding to less active or inactive genes.

The issue of biological repeats remains. Despite the original comments there are still places (eg Fig 3a and Fig 3h) in which standard deviation has been calculated from two biological repeats. This is not acceptable. Also, the number of repeats is not explicitly specified for Fig1b or Fig4g. It should be.

We have repeated ChIPs for Fig 3a, 3h, and Fig 4g with three biological replicates and added the new figures including the statistical analysis. ChIP for Fig 1b has been done with three biological replicates during the first round of revision. We have added this information in figure legend.

Reviewer #3 (Remarks to the Author):

Ha et al. have done a nice job addressing previous reviewer comments and improving this manuscript. In addition to better describing experimental methods, repetitions, and statistical tests, the authors have performed key experiments in response to reviewer suggestions. Importantly, the authors have repeated ChIP-seq experiments with inclusion of a spike-in control. This normalization step has changed one of the previous conclusions. Instead of *hda1Δ* reducing H3 occupancy, the authors now report a slight elevation in H3 occupancy, which correlates with the lack of cryptic initiation in these cells. Furthermore, the authors have now normalized the H4ac ChIP-seq data to total H3, bolstering their arguments that H4ac is significantly elevated on active coding regions in *hda1Δ* cells. Other valuable additions, which provide some mechanistic insights, are the experiments in Figures 3D and 4F. These two experiments probe the mechanism of Hda1C recruitment to coding regions and thus raise the overall impact of this study. Taken together, the new experiments and controls, improved description of experimental methods and reproducibility, and changes to the text have significantly improved this manuscript. The result is a strong paper that will be of interest to many in the chromatin/transcription fields.

We thank the reviewer for recognizing the significance of our findings and for the helpful comments.

I have a few minor suggestions:

1. Page 8--- I recommend changing "binds directly" to "associates with" in the header. ChIP assays don't show direct DNA binding.

The sentence has been changed.

2. Figure 5C and top of page 13--- Can the authors provide a brief explanation for the choice of STT4 and HAP1 for these ChIP experiments? Elsewhere, PMA1 and YEF3 were the test genes.

The effect of set2Δ on histone acetylation was relatively weak at PMA1 and YEF3. To see the synergistic effects of set2Δhda1Δ double deletion, we chose the two genes showing a significant increase of histone acetylation in mutants for Hda1 and Set2. We have added this information in the text.

3. Page 14, bottom--- The citations to Figures 2d and 2f are a bit confusing. Since those panels show normalized data, it's difficult to draw conclusions about total histone occupancy.

This has been corrected.

4. Figure 3 legend needs to be corrected. Panels h and i should be changed to g and h. Also, the sentence “The Arb2 domain is required for Hda1” is incomplete.

We thank the reviewer. These issues have been corrected.

5. Why are *K. lactis* strains mentioned in the strain list? I thought the spike-in strain was *S. pombe*.

The TRP1 gene is from K. lactis. The strain used for spike-in control has been added in the list.

REVIEWERS' COMMENTS:

Reviewer #2 (Remarks to the Author):

The authors have fully addressed all the issues that were raised by this reviewer.

Reviewer #2 (Remarks to the Author):

The authors have fully addressed all the issues that were raised by this reviewer.

We thank the reviewer for the comments.